# Relationship between simultaneously recorded spiking activity and fluorescence signal in GCaMP6 transgenic mice

Lawrence Huang[1†], Peter Ledochowitsch[1†], Ulf Knoblich[1], Jérôme Lecoq[1], Gabe J Murphy[1], R Clay Reid[1], Saskia EJ de Vries[1], Christof Koch[1], Hongkui Zeng[1*], Michael A Buice[1], Jack Waters[1*], Lu Li[1,2*]

[1]Allen Institute for Brain Science, Seattle, United States; [2]Guangdong Provincial Key Laboratory of Malignant Tumor Epigenetics and Gene Regulation, Guangdong-Hong Kong Joint Laboratory for RNA Medicine, Medical Research Center, Sun Yat-sen Memorial Hospital, Sun Yat-sen University, Guangzhou, China

**Abstract** Fluorescent calcium indicators are often used to investigate neural dynamics, but the relationship between fluorescence and action potentials (APs) remains unclear. Most APs can be detected when the soma almost fills the microscope's field of view, but calcium indicators are used to image populations of neurons, necessitating a large field of view, generating fewer photons per neuron, and compromising AP detection. Here, we characterized the AP-fluorescence transfer function in vivo for 48 layer 2/3 pyramidal neurons in primary visual cortex, with simultaneous calcium imaging and cell-attached recordings from transgenic mice expressing GCaMP6s or GCaMP6f. While most APs were detected under optimal conditions, under conditions typical of population imaging studies, only a minority of 1 AP and 2 AP events were detected (often <10% and ~20–30%, respectively), emphasizing the limits of AP detection under more realistic imaging conditions.

*For correspondence:
hongkuiz@alleninstitute.org (HZ);
jackw@alleninstitute.org (JW);
lilu67@mail.sysu.edu.cn (LL)

†These authors contributed equally to this work

Competing interests: The authors declare that no competing interests exist.

## Introduction

Genetically encoded calcium indicators (GECIs) are widely used with two-photon laser scanning microscopy to report neuronal activity within local populations in vivo (*Luo et al., 2018*). This optical approach is minimally invasive and enables simultaneous measurement of activity from hundreds or even thousands of neurons at single-cell resolution, over multiple sessions. Using a contemporary GECI such as GCaMP6s, fluorescence changes associated with isolated spikes (action potentials, APs) in vivo can be detected when imaged at sufficiently high spatiotemporal resolution (*Chen et al., 2013*) (http://dx.doi.org/10.6080/K02R3PMN). Yet undetected APs are common in population imaging experiments (*Theis et al., 2016*; *Berens et al., 2018*).

Inferring the underlying AP train or firing rate from calcium imaging remains challenging for several reasons. First, population imaging studies necessarily employ a large field of view containing many neurons. In contrast, the AP to calcium-dependent fluorescence transfer function is typically characterized with a soma filling the field of view of the microscope, to maximize photon flux from the soma and thereby signal-to-noise ratio. Second, there is no ground truth spiking information available for most neurons in a population. Spiking information, often from a cell-attached recording, can be used to refine the spike inference model and thereby optimize AP detection. Third, the AP to calcium-dependent fluorescence transfer function may be different for each neuron due to various intrinsic and extrinsic factors, such as neuron-to-neuron differences in indicator expression.

Compared to viral expression, transgenic mouse lines offer convenience (e.g. bypassing virus injection and associated procedures) and achieve more uniform GECI expression in genetically

**eLife digest** Neurons, the cells that make up the nervous system, transmit information using electrical signals known as action potentials or spikes. Studying the spiking patterns of neurons in the brain is essential to understand perception, memory, thought, and behaviour. One way to do that is by recording electrical activity with microelectrodes. Another way to study neuronal activity is by using molecules that change how they interact with light when calcium binds to them, since changes in calcium concentration can be indicative of neuronal spiking. That change can be observed with specialized microscopes know as two-photon fluorescence microscopes. Using calcium indicators, it is possible to simultaneously record hundreds or even thousands of neurons. However, calcium fluorescence and spikes do not translate one-to-one.

In order to interpret fluorescence data, it is important to understand the relationship between the fluorescence signals and the spikes associated with individual neurons. The only way to directly measure this relationship is by using calcium imaging and electrical recording simultaneously to record activity from the same neuron. However, this is extremely challenging experimentally, so this type of data is rare.

To shed some light on this, Huang, Ledochowitsch et al. used mice that had been genetically modified to produce a calcium indicator in neurons of the visual cortex and simultaneously obtained both fluorescence measurements and electrical recordings from these neurons. These experiments revealed that, while the majority of time periods containing multi-spike neural activity could be identified using calcium imaging microscopy, on average, less than 10% of isolated single spikes were detectable. This is an important caveat that researchers need to take into consideration when interpreting calcium imaging results.

These findings are intended to serve as a guide for interpreting calcium imaging studies that look at neurons in the mammalian brain at the population level. In addition, the data provided will be useful as a reference for the development of activity sensors, and to benchmark and improve computational approaches for detecting and predicting spikes.

defined neuronal populations (*Madisen et al., 2015*; *Daigle et al., 2018*). Using our intersectional transgenic mouse lines that enable Cre recombinase-dependent expression of GCaMP6s or GCaMP6f, we simultaneously characterized the spiking activity and fluorescence of individual GECI-expressing pyramidal neurons in layer 2/3 of mouse primary visual cortex (V1). We then tested the performance of several spike inference models, detecting APs under optimal conditions (models refined using the spiking information, with the soma filling the field of view) and under the less optimal conditions typical of population imaging experiments. Our results provide insight into the relationship between spiking activity in vivo and fluorescence signals and will aid the interpretation of existing and future calcium imaging datasets.

## Results

To characterize the single-cell transfer function between observed fluorescence signals and underlying APs in vivo, we performed simultaneous calcium imaging and cell-attached recordings in V1 L2/3 excitatory pyramidal neurons in anesthetized mice (*Figure 1A,B*). To directly compare our results to virally expressed GCaMP6f and GCaMP6s (*Chen et al., 2013*) (http://dx.doi.org/10.6080/K02R3PMN), we used a small field of view (19.3–27.3 × 19.3–21.5 μm; scanning rate 141.3–158.3 frames per second [fps]). 2–10 min recordings were obtained from 213 neurons, all with fluorescence excluded from the nucleus. Quality control code was deployed to exclude from further analysis recordings with artifacts such as motion, photobleaching, somatic dye from the recording pipette, electrophysiological or fluorescence baseline instability, and abrupt changes in AP waveform (see 'Materials and methods'). The dataset for further analysis was from 48 neurons from mice of four transgenic lines, two expressing GCaMP6s and two GCaMP6f in excitatory neurons in layer 2/3 and deeper layers of cortex (*Table 1*).

We analyzed events with fluorescence transients separated from those of adjacent events, containing a total of 5427 APs (28% of APs; *Figure 1—figure supplement 1C*). An event was defined as

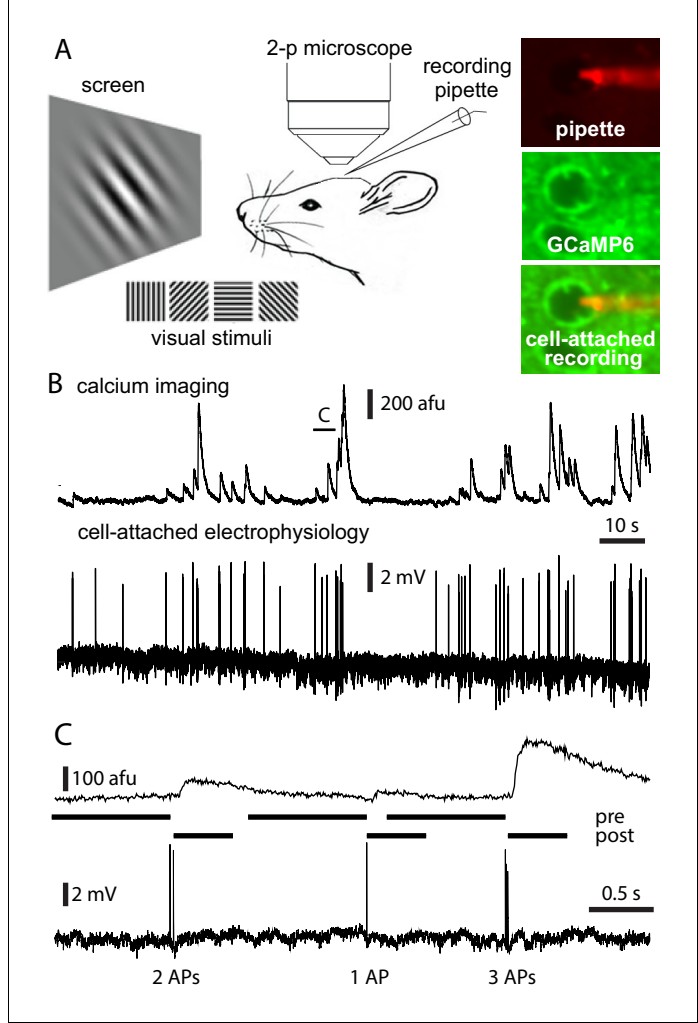

**Figure 1.** Simultaneous calcium imaging and electrophysiology in vivo. (**A**) Experimental design. (**B**) Fluorescence and Vm traces from an exemplar Emx1-s neuron. (**C**) 5 s of data from the neuron in panel B, showing a 2 AP, a 1 AP, and a 3 AP event. Pre- and post-AP exclusion windows, used to separate events, are illustrated for each event. AP, action potential.

The online version of this article includes the following figure supplement(s) for figure 1:

**Figure supplement 1.** Mouse age, cell depth, and firing rate.

one or more APs within 250 ms, with no APs in the preceding or subsequent 300 ms for GCaMP6f or 1 s and 500 ms for GCaMP6s (*Figure 1C*).

**Table 1.** Dataset.

Sample size for each mouse line (see also *Figure 1—figure supplement 1*).

| Mouse line | Acronym | GECI | Mice | Cells | Recording duration | APs per neuron |
|---|---|---|---|---|---|---|
| Emx1-IRES-Cre; Camk2a-tTA;Ai94 | Emx1-s | GCaMP6s | 5 | 21 | 241 ± 32 s | 478 ± 121 |
| Camk2a-tTA; tetO-GCaMP6s | tetO-s | GCaMP6s | 1 | 4 | 347 ± 108 s | 348 ± 71 |
| Cux2-CreERT2; Camk2a-tTA;Ai93 | Cux2-f | GCaMP6f | 3 | 12 | 300 ± 79 s | 484 ± 112 |
| Emx1-IRES-Cre; Camk2a-tTA;Ai93 | Emx1-f | GCaMP6f | 4 | 11 | 201 ± 23 s | 219 ± 38 |

## Calcium transients differ across mouse lines

Fluorescence measured from the soma is contaminated with fluorescence from the surrounding neuropil, due to the extended nature of the microscope point spread function. Neuropil contamination is often removed by subtracting a scaled version of the neuropil fluorescence from the somatic fluorescence, with the scale factor referred to as the r value (*Kerlin et al., 2010*; *Akerboom et al., 2012*). The r value can affect AP detection, with under-subtraction of neuropil leading to false positives (events detected when there was activity in the neuropil but not the soma) and over-subtraction leading to false negatives (failure to detect somatic activity). We examined the effects of r on detection of 1 AP events, with electrical recordings providing ground truth (*Figure 2* and *Figure 2—figure supplement 1*). For many GCaMP6s neurons, the receiver operating characteristic (ROC) curve changed little with r (*Figure 2A*), indicating that APs were detected with few false positives with little effect of neuropil subtraction. Neuropil subtraction exerted a stronger influence on event detection in GCaMP6f neurons, where the ROC curve changed with r (*Figure 2A*), permitting identification of the optimal r as that which maximized the area under the ROC curve and, thereby, the true/false event detection ratio. Optimal r for Emx1-f and Cux2-f neurons was approximately normally distributed with mean ± SEM of 0.82 ± 0.07 (20 neurons, *Figure 2B*). Our results indicate that the value of r has a modest effect on event detection in GCaMP6f neurons in mouse V1. The effect of neuropil subtraction may be greater during coordinated activity across the whole network, such as during strong sensory stimuli.

After neuropil subtraction (see 'Materials and methods'), we averaged trials by number of APs, fit a sum of exponentials to estimate rise and decay time constants and calculated peak ΔF/F (mean fluorescence over 100 ms around the maximum within 300 ms for GCaMP6f and 500 ms for GCaMP6s) for events with 1–5 APs (*Figure 3*). 28–55% of detected APs were in events with 1–5 APs (*Figure 1—figure supplement 1*) and >70% of these analyzed APs were in multi-AP events. As expected, peak ΔF/F increased approximately linearly with 1–5 APs, and peak ΔF/F and decay time constant were greater with GCaMP6s than GCaMP6f (*Figure 3C*). Peak ΔF/F was comparable to or slightly greater than in previous studies with GCaMP6s and GCaMP6f, possibly because we subtracted more of the neuropil fluorescence with a slightly greater r value (r = 0.8 vs. 0.7; *Chen et al., 2013*).

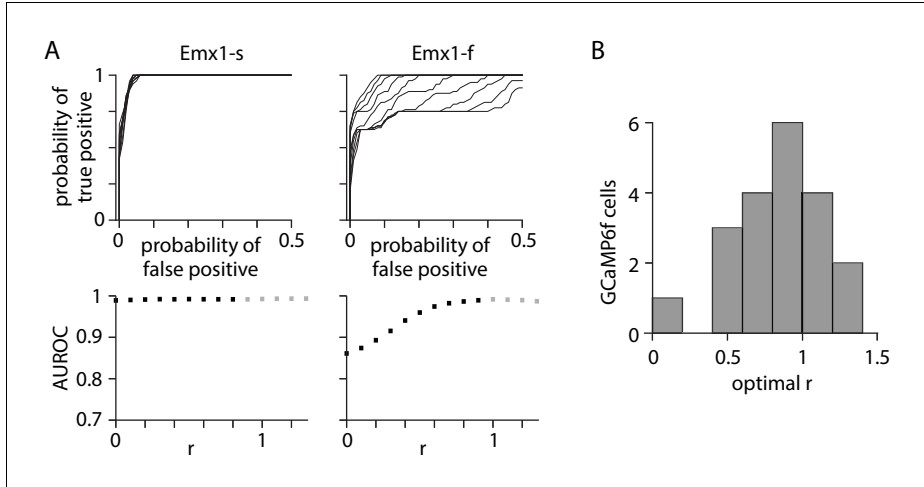

**Figure 2.** Neuropil subtraction optimized for 1 AP (action potential) detection. (**A**) Effect of changing neuropil subtraction on detection for exemplar GCaMP6s and GCaMP6f neurons. Upper plots: family of receiver operating characteristic (ROC) curves. Each curve illustrates detection probability for true APs against probability of false positives as detection threshold is changed, for 1 AP events. False positives were calculated from time windows with no APs. Each ROC curve represents a different value of r. Lower plots: area under the ROC curve as a function of r. Gray symbols represent value of r for which r * $F_{neuropil}$(t) was greater than $F_{cell\_measured}$(t), resulting in a negative F0 and inversion of the ΔF/F trace. (**B**) Distribution of r values for 20 GCaMP6f neurons.

The online version of this article includes the following figure supplement(s) for figure 2:

**Figure supplement 1.** Simulated effect of neuropil subtraction on event detection.

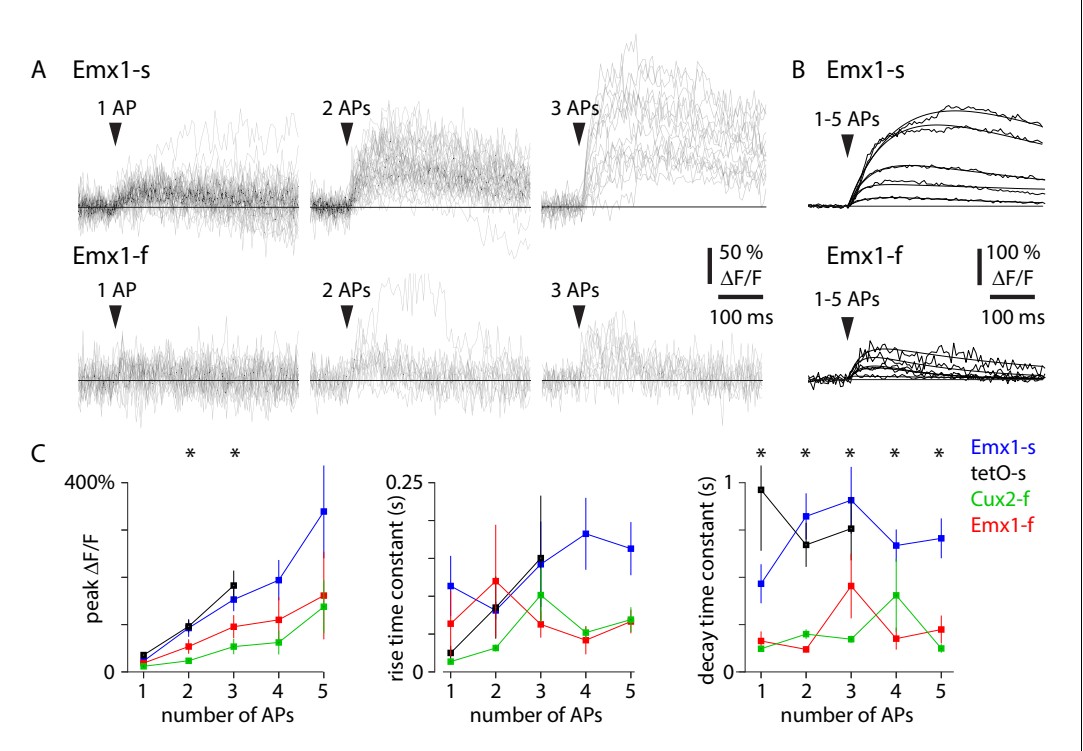

**Figure 3.** Action potential (AP)-evoked calcium transients in four mouse lines. (**A**) Example fluorescence traces (ΔF/F, with r = 0.82 neuropil subtraction) for 1 AP, 2 AP, and 3 AP events for an exemplar Emx1-s neuron (forty-two 1 AP events, thirty-eight 2 AP events, sixteen 3 AP events) and an exemplar Emx1-f neuron (twenty-three 1 AP events, twelve 2 AP events, eleven 3 AP events). (**B**) Mean fluorescence traces and fits (sum of two exponentials) for 1–5 AP events for the two neurons in A. (**C**) Mean ± SEM peak DF/F, rise time constant, and decay time constant for four mouse lines. Number of neurons for 1–5 AP events were: 15, 16, 14, 9, 6 for Emx1-s; 4, 4, 4, 0, 0 for tetO-s; 10, 10, 10, 5, 4 for Emx1-f; 9, 9, 7, 4, 2 for Cux2-f. Asterisks indicate differences between mouse lines (p<0.05, one-way ANOVA).

The online version of this article includes the following figure supplement(s) for figure 3:

**Figure supplement 1.** Trial-to-trial variability of 1 AP (action potential) events.

As expected, photon shot noise was the dominant noise source in images from all mouse lines. The pixelwise slope of the least squares fit between the variance and mean of the photon flux was 1.04 ± 0.01 (mean ± SEM), consistent with the noise following a Poisson process (intercept −0.08 ± 0.2, 48 neurons). Trial-to-trial variability in the amplitude of the 1 AP-evoked fluorescence was substantial and exceeded photon shot noise in most neurons (*Figure 3—figure supplement 1*). The sources of non-Poisson variability in our results are unclear, but negligible motion was visible in the movies after motion correction. Likely the variability results primarily from trial-to-trial differences in the AP-evoked calcium concentration, assuming GCaMP6f and GCaMP6s are expressed at sufficient concentrations to report resting changes in calcium concentration in all four mouse lines.

## Increasing field of view reduces optimized event detection

GCaMP6 indicators have been widely adopted because they exhibit greater AP-evoked ΔF/F than previous GCaMP indicators, but still some APs may go undetected (*Chen et al., 2013*). Under ideal conditions, almost all APs can be detected (with probability close to 1 at a false positive probability of 1%; *Chen et al., 2013*). However, many imaging experiments are performed with a field of view of hundreds of micrometers and this large field of view limits the dwell time per soma and thereby the photon flux per soma and signal-to-noise ratio. What event detection rate might be expected when imaging a large field of view, sufficient to include hundreds of somata? How much does field of view affect event detection?

We calculated detection probability for 1 AP and 2 AP events, using AP times from electrophysiology recordings to optimize event detection for each neuron (*Chen et al., 2013*). In high spatial

and temporal resolution images, the probability of 1 AP event detected spanned a wide range (probability 0.07–1 and 0.11–0.95 for GCaMP6s and GCaMP6f, at 1% false positive probability, *Figure 4A-C*). As expected (*Dana et al., 2014*; *Wei et al., 2019*), most 1 AP events were detected in GCaMP6s and GCaMP6f neurons, but with lower average probability in GCaMP6f neurons (1 AP detection probability 0.70 ± 0.06 for 18 Emx1-s neurons, 0.80 ± 0.03 for three tetO-s neurons, 0.40 ± 0.08 for nine Cux2-f neurons, 0.60 ± 0.08 for 11 Emx1-f neurons, mean ± SEM at 1% false positive probability). 2 AP events were reliably detected in all four mouse lines (*Figure 4C*; detection probability 0.90 ± 0.06 for Emx1-s, 1.0 ± 0.0 for tetO-s, 0.66 ± 0.07 for Cux2-f, 0.80 ± 0.05 for Emx1-f, at 1% false positive probability). In high spatial and temporal resolution images, in all four mouse

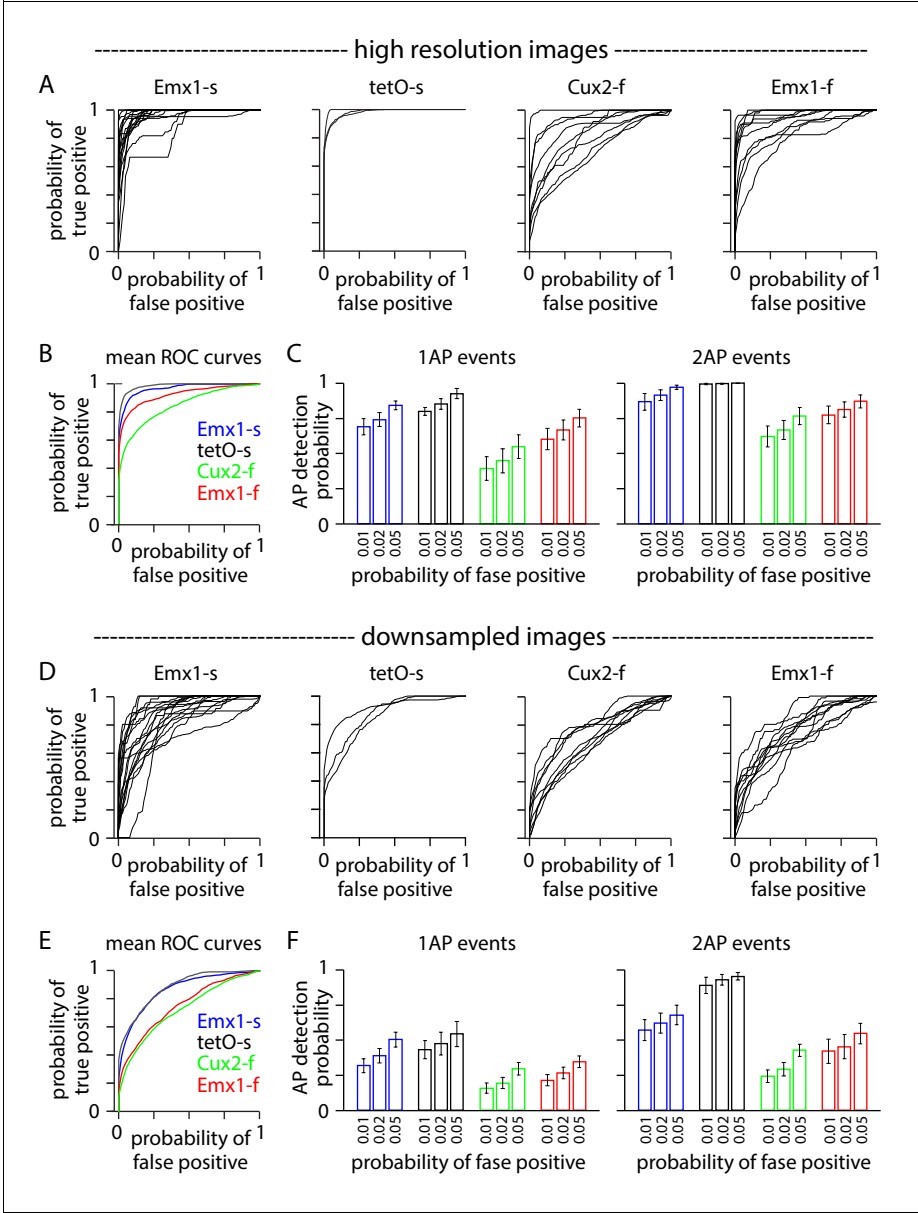

**Figure 4.** Downsampling affects event detection. (**A–C**) Event detection from images at high spatial and temporal resolution. (**A**) Receiver operating characteristic (ROC) curves for 1 AP events in 42 neurons, organized by mouse line. Neuropil subtraction was performed with r = 0.8 where possible (see 'Materials and methods'). Numbers of neurons were 18 Emx1-s, 3 tetO-s, 11 Cux2-f, 9 Emx1-f. (**B**) Mean ROC curves for the four mouse lines. (**C**) Event detection probabilities for 1 AP and 2 AP events. Bars represent mean ± SEM. (**D–F**) Equivalent plots for the same neurons after downsampling.

lines, it was possible to detect most but by no means all 1 AP and 2 AP events with a false positive probability of only 1%.

In these transgenic mice, 1 AP detection probabilities were lower than in previously reported neurons with virally expressed GCaMP6s and GCaMP6f (0.99 and 0.84 at 1% false positive probability in *Chen et al., 2013*). There are several possible reasons for this difference. In the transgenic mice used here, GCaMP expression is widespread throughout neocortex, which may result in labeling of greater numbers of axons and dendrites that contribute to the neuropil signal. Furthermore, GCaMP6 expression may be weaker in the four TIGRE1.0 mouse line crosses examined here than with strong promoter-driven adeno-associated virus (AAV) vectors as used in *Chen et al., 2013*. The newer TIGRE2.0 reporter lines drive GCaMP expression that is comparable to that from strong promoter-driven AAVs (*Daigle et al., 2018*), likely enabling 1 AP and 2 AP detection rates in transgenic mice that are comparable to those achieved with viral expression of GCaMP6.

Our recordings were obtained with a small field of view, at a high frame rate and centered on the soma (~19.3×19.3 μm, ~158 Hz, *Figure 5A,B*). In an attempt to simulate commonly used imaging conditions, we downsampled our images in space and time to mimic imaging with a 412 × 412 μm field of view at 30.3 Hz, as used in the Allen Brain Observatory (*Figure 5C,D*). The baseline fluorescence noise from downsampled images was comparable to that in the Allen Brain Observatory (*Figure 5E*) and is presumably comparable to images in many two-photon datasets with populations of hundreds of neurons.

As expected, event detection probabilities were lower for downsampled images than for the original, high-resolution images (*Figure 4D–F*). 1 AP and 2 AP event detection probabilities were 0.32 ± 0.05 and 0.55 ± 0.08 for 18 Emx1-s neurons, 0.43 ± 0.07 and 0.89 ± 0.06 for three tetO-s neurons, 0.16 ± 0.04 and 0.24 ± 0.04 for 9 Cux2-f neurons, 0.21 ± 0.04 and 0.42 ± 0.09 for 11 Emx1-f neurons (mean ± SEM at 1% false positive probability). Even for 2 AP events, detection probability is <0.5 when imaging with GCaMP6f and a field of view of several hundred micrometers.

In summary, 1 AP and 2 AP events were detected with high probability when images were acquired with high spatial and temporal resolution and when analysis was performed with an event detection algorithm optimized for each neuron using known AP times. Even with known AP times to optimize detection for each neuron, event detection was severely impaired by a reduction in spatial and temporal resolution to

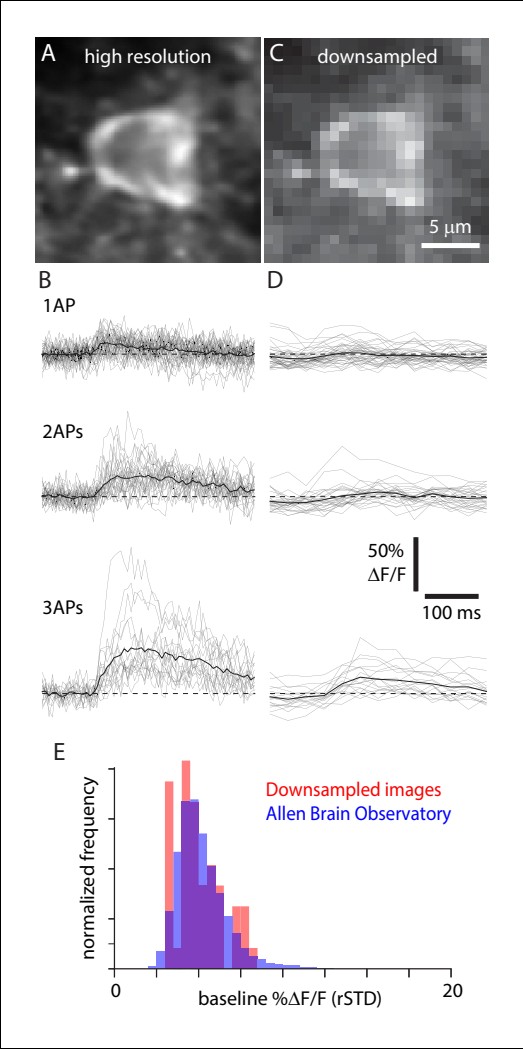

**Figure 5.** Downsampling mimics large field-of-view images. (**A**) Original, high spatial and temporal resolution GCaMP6f image of an exemplar Cux2-f neuron. (**B**) 1 AP, 2 AP, and 3 AP traces for the same neuron. Thin lines, individual trials; thick line, mean. Dashed line, ΔF/F = 0. (**C**) Downsampled image of the same neuron. (**D**) Traces from the downsampled neuron. (**E**) Normalized distribution of baseline noise (robust standard deviation [rSTD]) for 48 downsampled neurons (red) and 11,816 layer 2/3 neurons from Emx1-f and Cux2-f mice in the Allen Brain Observatory (blue). The online version of this article includes the following figure supplement(s) for figure 5:

**Figure supplement 1.** Example fluorescence traces from image quality control procedures, implemented during image downsampling.

mimic a typical two-photon population imaging experiment.

## Modest effect of field of view on event detection under typical imaging conditions

In a typical two-photon population imaging experiment, no electrophysiology recording is available to optimize event detection. Often, the shape of the calcium transient is estimated from published indicator rise and decay times or derived from a representative sample of fluorescence transients. What are typical event detection and false positive probabilities under these sub-optimal conditions when the underlying AP times are unknown? Is performance degraded equally for high and for low resolution images?

We compared event detection from high-resolution and downsampled images using three spike inference algorithms: unconstrained non-negative deconvolution (NND; *Friedrich et al., 2017*), non-negative deconvolution with an L0 constraint to enforce event sparseness (Exact L0; *Jewell et al., 2020*), and a biophysical model that explicitly accounts for intracellular calcium dynamics (MLspike; *Deneux et al., 2016*). These three algorithms are among the highest performing spike inference algorithms (*Berens et al., 2018*).

For each neuron, the algorithms were deployed to estimate the number of APs in each image of the movie. All three algorithms estimated AP numbers that approximately recapitulated the number of APs measured with electrophysiology, but the number of APs per frame was typically not an integer due to imperfect spike inference (*Figure 6A,D*). We characterized performance using the Pearson correlation coefficient and the Matthews correlation coefficient, which compare measured and estimated AP number at each time point and the presence or absence of an event at each time point, respectively. Pearson correlation coefficients were ~0.4 when calculated with 33 ms time bins, increasing toward 0.7 as bin size was increased to 500 ms (*Figure 6—figure supplement 1*), comparable to published results (*Berens et al., 2018*). Mean Pearson and Matthews correlation coefficients were similar across inference algorithms and mouse lines and differed little between high-resolution and downsampled images (*Figure 6B,E*).

We plot ROC curves to more directly examine the relationship between detected events and false positives. Since spike inference is generally useful only where false positive rates are low, we focused on false positive probabilities in the range of 0–0.05. Performance differed greatly between neurons, but mean ROC curves were similar across mouse lines, with only modest differences between algorithms, between GCaMP6s and GCaMP6f lines, or between high-resolution and downsampled conditions (*Figure 6C,F*).

Naturally, detection probability increased with the number of APs per event. At a false positive probability of 0.01, detection probability was commonly <0.1 for 1 AP events, increasing approximately linearly with AP number, often to ~0.8 for 5 AP events (*Figure 7A,B*). With 1 AP events being the most common event type in all four mouse lines (*Figure 7C*), it was possible to detect only a minority of events with a low false positive probability. Using these spike inference algorithms, although detection probabilities were commonly slightly lower for downsampled than for high-resolution images, the difference was modest, indicating that the decreased SNR of population imaging had little effect on event detection in our dataset.

Using our dataset, we compared event detection with three algorithms: unpenalized NND, NND with L0 constraint and mathematically guaranteed globally optimal solution (Exact L0), and the biophysically inspired MLspike model. For NND, performance was poor at 30 Hz and considerably improved by upsampling to 150 Hz (*Figure 7—figure supplement 1*). Upsampling of low frame rate data, often to 100 Hz, is a common practice in the field (*Theis et al., 2016*; *Berens et al., 2018*; *Pachitariu et al., 2018*). For MLspike, performance was poor without use of the autocalibration procedure to optimize the model for each neuron (*Figure 7—figure supplement 2*). MLspike thus contrasted with deconvolution-based algorithms, for which fixed parameters are more effective (*Pachitariu et al., 2018*). For Exact L0, neither upsampling nor optimization for each neuron was necessary for optimal mean performance across neurons.

*Pachitariu et al., 2018* found that unpenalized NND matched and often exceeded the performance of algorithms with sparsifying constraints such as NND with an approximate L0 constraint (no mathematical guarantee of globally optimal solution). Consistent with the conclusions of *Pachitariu et al., 2018*, Exact L0 lagged the performance of NND for some metrics and genotypes (*Figure 7* A, B) but was indistinguishable or even superior for others (*Figure 6B, E*). The

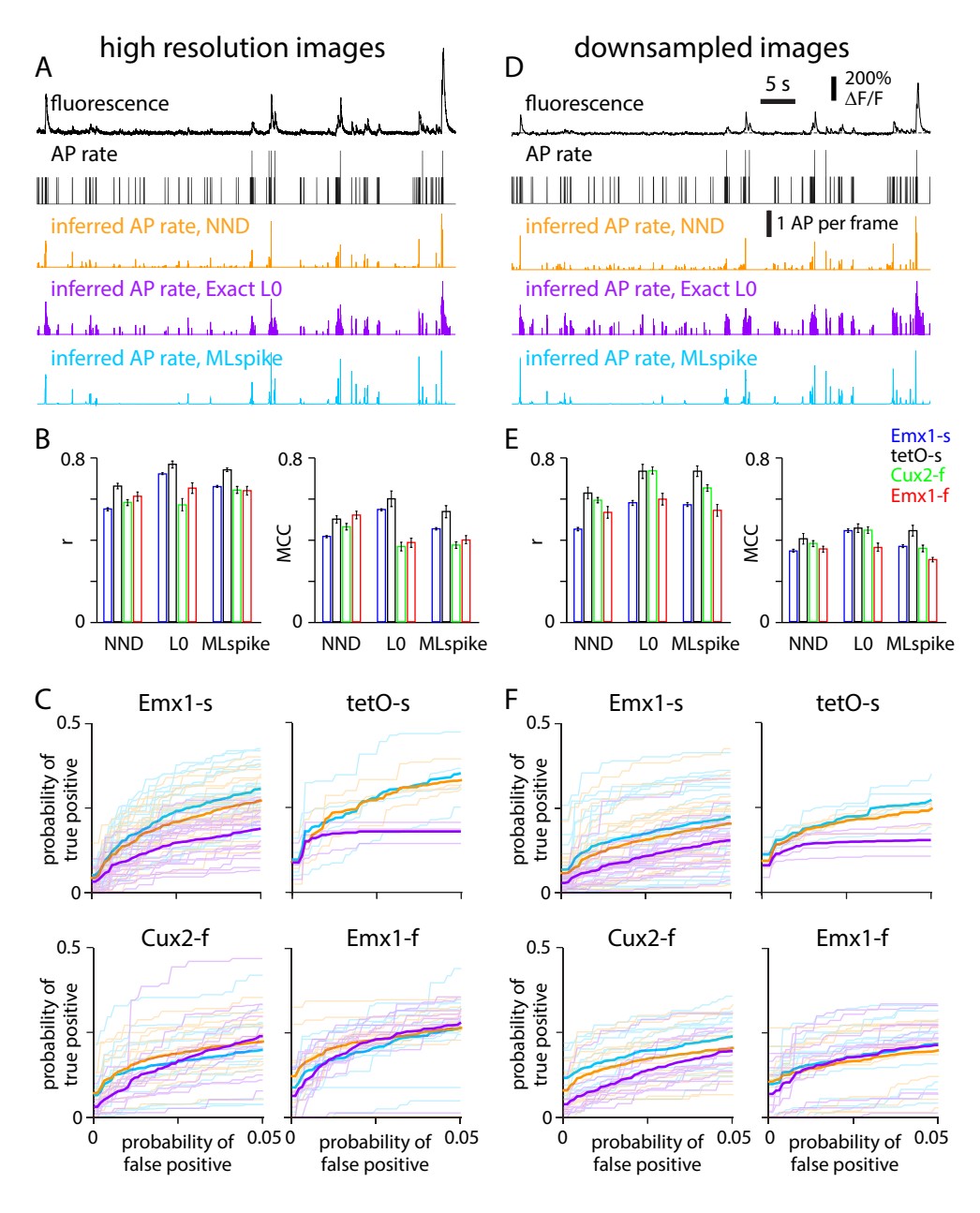

**Figure 6.** Performance of spike inference algorithms on high-resolution and downsampled images. (**A**) Results from an exemplar Cux2-f neuron, at high resolution. Fluorescence and action potential (AP) rate from electrophysiology (black). Below, APs per image frame estimated with three spike inference algorithms: MLspike (blue), Exact L0 (purple), and non-negative deconvolution (NND, orange). (**B**) Pearson correlation coefficient (r) and Matthews correlation coefficient (MCC) for the three algorithms for each mouse line. 300 ms bins. (**C**) Receiver operating characteristic (ROC) curves, reporting probabilities of detecting true and false events in each time bin. Thin lines: individual neurons. Thick lines: mean across neurons. 300 ms bins. (**D–F**) Equivalent plots for downsampled images.

The online version of this article includes the following figure supplement(s) for figure 6:

**Figure supplement 1.** Effect of bin duration on measures of spike inference algorithm performance.

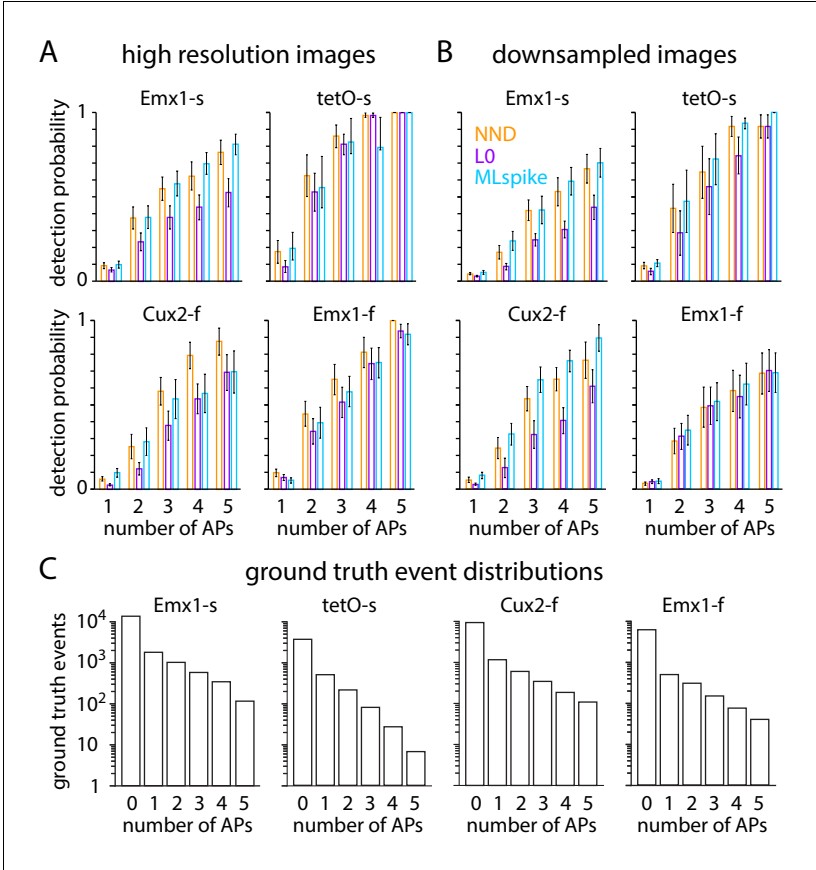

**Figure 7.** Performance of blind spike inference algorithms for 1–5 AP (action potential) events. (**A**) Mean ± SEM detection probabilities at 1% false positive probability for high-resolution images. (**B**) Mean ± SEM detection probabilities at 1% false positive probability for downsampled images. (**C**) Frequency of 0–5 AP events for 250 ms bins for each mouse line.

The online version of this article includes the following figure supplement(s) for figure 7:

**Figure supplement 1.** Upsampling enhances performance of unconstrained non-negative deconvolution (NND).

**Figure supplement 2.** Autocalibration enhanced performance of MLspike.

**Figure supplement 3.** Comparison of spike inference with blind and ground truth action potential (AP)-optimized algorithms.

performance of MLspike was broadly equivalent to that of NND, but the loss of performance due to downsampling was less with MLspike, resulting in outperformance of MLspike on downsampled images. Our results point to MLspike as a compelling choice for spike inference in population imaging experiments. Our results also suggest that there is ample room for improvement of spike inference models since event detection by the three spike inference models falls far short of the performance of the ground truth-optimized approach employed in *Figure 4* (*Figure 7—figure supplement 3*).

In summary, relative to small field-of-view imaging, population imaging conditions decreased the probability of spike event detection with an event detector optimized to each individual neuron using ground truth AP information (*Figure 4*). With blind spike inference, many events went undetected even under near-ideal imaging conditions with a small field of view, and event detection was not substantially worse under population imaging conditions (*Figure 6*, *Figure 7*, *Figure 7—figure supplement 3*). The results of *Figures 6* and *7* are likely representative of event detection in many GCaMP6 imaging experiments, where ground truth AP information is not available and blind spike inference is employed. Our results indicate that even though GCaMP6 indicators are bright and sensitive enough to enable the detection of most 1 AP events in superficial cortical pyramidal neurons in vivo if the detection procedure is optimized using ground truth AP information, most events

containing 1, 2, and sometimes greater numbers of APs go undetected in our (and likely in many other) imaging experiments with GCaMP6.

## Discussion

Calcium imaging is widely used to report neuronal spiking activity in vivo. However, accurate spike inference from calcium imaging remains a challenge, and there are relatively few ground truth datasets with simultaneous calcium imaging and electrophysiology to aid the development of more accurate spike inference algorithms. In a recent challenge, ~40 algorithms were trained and tested on datasets consisting of 37 GCaMP6-expressing neurons, underscoring the need for additional GCaMP6 calibration data (*Berens et al., 2018*). In addition to supporting efforts toward spike inference, an improved understanding of the relationship between spiking and observed fluorescence signals is necessary to further broaden the utility and impact of calcium imaging. To these ends, we contribute a ground truth dataset consisting of 48 V1 L2/3 excitatory neurons recorded at single-cell resolution (available at https://portal.brain-map.org/explore/circuits/oephys) and characterized their AP-to-calcium fluorescence transfer function. Complementing existing datasets with viral GECI expression (*Chen et al., 2013*; *Theis et al., 2016*; *Dana et al., 2016*), our work facilitates interpretation of existing and future calcium imaging studies using mainstream transgenic mouse lines, such as the Allen Institute's Brain Observatory Visual Coding dataset (http://observatory.brain-map.org/visualcoding) (*de Vries et al., 2020*).

Previous studies have established that most APs can be detected with GCaMP6 indicators under near-optimal conditions (*Chen et al., 2013*). Yet undetected APs are common in population imaging experiments (*Theis et al., 2016*; *Berens et al., 2018*). To investigate why APs are often missed during population imaging, we compared event detection in 250 ms time windows with a neuronal soma occupying most of the image, near-optimal conditions for AP event detection, and event detection when the soma occupies just a small percentage of the field of view, less ideal conditions that are common in population imaging studies. Importantly, we downsampled images to simulate population imaging conditions, enabling comparison for the same APs under different imaging conditions.

Our results indicate that, in GCaMP6 transgenic mice, most APs can be detected under near-optimal conditions, while detection is less effective during population imaging. These conclusions are similar to those of previous studies with viral GCaMP6 expression, but our results also reveal two reasons for the difference in detection. Unsurprisingly, the reduced signal-to-noise ratio of population imaging, relative to single soma imaging, results in less effective event detection. However, a high signal-to-noise ratio, achieved by imaging one soma, was no guarantee of effective event detection. Effective detection also required optimization of detection for the neuron of interest, using known AP times to identify events with different AP numbers and so generate kernels of the appropriate amplitude and time course. Parameter tuning in the absence of known AP times, with the MLspike autocalibration routine, improved event detection but not to the high standard of ground truth-optimized detection. Unfortunately, measuring AP times for every neuron with electrophysiology is rarely feasible, severely limiting the percentage of events one might reasonably expect to detect with GCaMP6 in most imaging experiments.

Our results point to several practices that might be adopted to maximize spike detection. First, minimize the field of view, hence maximizing photon flux per neuron. Second, tune the spike inference model for each neuron independently, where possible. Third, compare the results of several spike inference models. The three models employed here produced similar AP detection rates, whether applied to high-resolution or to downsampled images. Similarly, *Pachitariu et al., 2018* observed that the L0 constraint failed to improve performance of the NND model. Nonetheless, each model has strengths and weaknesses. For example, a model may detect more APs than another but at the cost of a greater false positive rate. As a result, model performance may diverge for some AP rates and patterns. In the worst case, comparing models provides some protection from errors in implementation. Fourth, ensure that traces are sampled (or upsampled) at a sufficiently high rate when employing NND and use autocalibration with MLspike; both make a substantial difference to model performance. Finally, exercise caution when interpreting the inferred spike rates. Commonly, many APs are not detected using even the most accurate spike inference models.

In summary, in this study we present a ground truth dataset from anesthetized mice with simultaneous electrophysiology and calcium imaging. Analysis of this dataset revealed that only a small fraction of isolated APs were detected under typical population imaging conditions and with existing spike inference algorithms. By making our data freely available, we hope that it will serve the community as a further resource to better understand the quantitative link between calcium-evoked fluorescent imaging signals and spiking activity.

## Materials and methods

### Key resources table

| Reagent type (species) or resource | Designation | Source or reference | Identifiers | Additional information |
|---|---|---|---|---|
| Genetic reagent *Mus musculus* | B6.129S2-Emx1tm1(cre)Krj/J, Emx1-IRES-Cre | Jackson Laboratory | RRID:IMSR_JAX:005628 RRID:MGI:2684610 | |
| Genetic reagent *Mus musculus* | B6(Cg)-Cux2tm3.1 (cre/ERT2)Mull/Mmmh, Cux2-CreERT2 | MMRRC | RRID:MMRRC_032779-MU RRID:MGI:5014172 | |
| Genetic reagent *Mus musculus* | B6.Cg-Tg(Camk2a-tTA) 1Mmay/DboJ, Camk2a-tTA | Jackson Laboratory | RRID:IMSR_JAX:007004 RRID:MGI:2179066 | |
| Genetic reagent *Mus musculus* | B6;DBA-Tg(tetO-GCaMP6s) 2Niell/J, tetO-GCaMP6s | Jackson Laboratory | RRID:IMSR_JAX:024742 RRID:MGI:5553332 | |
| Genetic reagent *Mus musculus* | B6;129S6-Igs7tm93.1 (tetO-GCaMP6f)Hze/J, Ai93(TITL-GCaMP6f) | Jackson Laboratory | RRID:IMSR_JAX:024103 RRID:MGI:5558086 | |
| Genetic reagent *Mus musculus* | B6.Cg-Igs7tm94.1 (tetO-GCaMP6s)Hze/J, Ai94(TITL-GCaMP6s) | Jackson Laboratory | RRID:IMSR_JAX:024104 RRID:MGI:5607576 | |
| Software, algorithm | MATLAB R2016b | http://www.mathworks.com/products/matlab/ | RRID:SCR_001622 | |
| Software, algorithm | Python 3.7.4 | http://www.python.org/ | RRID:SCR_008394 | |
| Software, algorithm | LabVIEW 2015 | http://www.ni.com/labview/ | RRID:SCR_014325 | |

Experimental procedures were conducted in accordance with NIH guidelines and approved by the Institutional Animal Care and Use Committee (IACUC) of the Allen Institute for Brain Science under protocol number 1509.

### Mice

Two-photon-targeted electrophysiology and two-photon calcium imaging were conducted in 2- to 5-month-old male and female transgenic mice: five *Emx1-IRES-Cre;Camk2a-tTA;Ai94* (Emx1-s) mice, one *Camk2a-tTA;tetO-GCaMP6s* (tetO-s) mouse, three *Emx1-IRES-Cre;Camk2a-tTA;Ai93* (Emx1-f) mice, and four *Cux2-CreERT2;Camk2a-tTA;Ai93* (Cux2-f) mice. All four lines drive GCaMP expression primarily in excitatory neurons. In Cux2-CreERT2 mice, Cre and GCaMP expression are enriched in layer 2/3 (*Franco et al., 2012*; *Harris et al., 2014*). In Emx1-IRES-Cre and Camk2a-tTA mice, GCaMP is expressed throughout cortical layers (*Gorski et al., 2002*; *Wekselblatt et al., 2016*). Images showing the pattern of Cre and GCaMP expression in these mouse lines are available via the Transgenic Characterization pages of the Allen Mouse Brain Connectivity Atlas and Allen Brain Observatory: https://connectivity.brain-map.org/transgenic, http://observatory.brain-map.org/visualcoding/transgenic.

Mice of some of the genotypes used here, most notably Emx1-f, can exhibit epileptiform activity (*Steinmetz et al., 2017*), including overt seizures. Mice with seizures were excluded from the study. However, the spiking patterns of neurons from GCaMP6s and -f lines commonly differed, suggesting that one or more transgenes affected cell or circuit activity (*Figure 1—figure supplement 1C*).

## Surgery

Mice were anesthetized with either isoflurane (0.75–1.5% in $O_2$) or urethane (1.5 g/kg, 30% aqueous solution, intraperitoneal injection), then implanted with a metal head-post. A circular craniotomy was performed with skull thinning over the left V1 centering on 1.3 mm anterior and 2.6 mm lateral to the lambda. During surgery, the craniotomy was filled with artificial cerebrospinal fluid (ACSF) containing (in mM): NaCl 126, KCl 2.5, $NaH_2PO_4$ 1.25, $MgCl_2$ 1, $NaHCO_3$ 26, glucose 10, $CaCl_2$ 2, in $ddH_2O$; 290 mOsm; pH was adjusted to 7.3 with NaOH to keep the exposed V1 region from overheating or drying. Durotomy was performed to expose V1 regions of interest (ROIs) that were free of major blood vessels to facilitate the penetration of recording micropipettes. A thin layer of low-melting-point agarose (1–1.3% in ACSF, Sigma-Aldrich) was then applied to the craniotomy to control brain motion. The mouse body temperature was maintained at 37°C with a feedback-controlled animal heating pad (Harvard Apparatus).

## Calcium imaging

Individual GCaMP6+ neurons ~100–300 μm below the pial surface of cortex were visualized under adequate anesthesia (stage III-3) using a Bruker (Prairie) two-photon microscope and Chameleon Ultra II Ti:sapphire laser (Coherent). Fluorescence excited at 920 nm wavelength, with <70 mW laser power measured after the objective, was collected in two spectral channels using green (510/42 nm) and red (641/75 nm) emission filters (Semrock) to visualize GCaMP6 and the Alexa Fluor 594-containing micropipette, respectively. Fluorescence images of 96–136 × 96–107 pixels and a 19.3–27.3 × 19.3–21.5 μm field of view were acquired at 141.3–158 frames per second through a 16× water immersion objective lens (Nikon, NA 0.8). Recordings included periods with and without visual stimuli. Mean ± SEM number of pixels per neuron was 1136 ± 46.

## Electrophysiology

Two-photon-targeted cell-attached recording was performed following established protocols (*Margrie et al., 2003*; *Kitamura et al., 2008*; *Knoblich et al., 2019*). Long-shank borosilicate (KG-33, King Precision Glass) micropipettes (5–10 MΩ) were pulled with a P-97 puller (Sutter) and filled with ACSF and Alexa Fluor 594 to perform cell-attached recordings on GCaMP6+ neurons. Micropipettes were installed on a MultiClamp 700B headstage (Molecular Devices), which was mounted onto a Patchstar micromanipulator (Scientifica) with an approaching angle of 31° from horizontal plane. Minimal seal resistance was 20 MΩ. Data were acquired under 'I = 0' mode (zero current injection) with a Multiclamp 700B, recorded at 40 kHz using Multifunction I/O Devices (National Instruments) and custom software written in LabVIEW (National Instruments) and MATLAB (MathWorks). Isoflurane level was intentionally adjusted during recording sessions to keep the anesthesia depth as light as possible, resulting in fluctuation of the firing rates of recorded neurons.

## Visual stimulation

Whole-screen sinusoidal static and drifting gratings were presented on a calibrated LCD monitor spanning 60° in elevation and 130° in azimuth to the contralateral eye. The mouse's eye was positioned ~22 cm away from the center of the monitor. For static gratings, the stimulus consisted of four orientations (45° increment), four spatial frequencies (0.02, 0.04, 0.08, and 0.16 cycles per degree), and four phases (0, 0.25, 0.5, 0.75) at 80% contrast in a random sequence with 10 repetitions. Each static grating was presented for 0.25 s, with no inter-stimulus interval. A gray screen at mean illuminance was presented randomly a total of 60 times. For drifting gratings, the stimulus consisted of eight orientations (45° increment), four spatial frequency (0.02, 0.04, 0.08, and 0.16 cycles per degree), and one temporal frequency (2 Hz), at 80% contrast in a random sequence with up to five repetitions. Each drifting grating lasted for 2 s with an inter-stimulus interval of 2 s. A gray screen at mean illuminance was presented randomly for up to 15 times.

## Neuron selection

We obtained recordings from 213 neurons and developed a numerical routine to exclude neurons with questionable electrophysiology or fluorescence movies, such as abrupt changes in baseline voltage or AP waveform or image artifacts such as those due to motion, photobleaching, or other slow baseline changes. Neurons were accepted for analysis if they passed both electrophysiology and

image quality control criteria. Electrophysiology quality control is described in the next section and imaging quality control in the 'Image downsampling' section; 145 and 10 neurons were eliminated in the electrophysiology and image quality control steps, leaving 58 neurons. Of these, 10 were excluded from further analysis: red indicator had entered the soma from the pipette in seven instances, two neurons segmented poorly during image analysis, and one had a truncated electrophysiology recording. The final dataset consisted of 48 neurons.

## Electrophysiology quality control

Electrophysiology traces were first baseline-subtracted to remove slow drift (third-order Savitzky-Golay filter over 20,001 samples using MATLAB sgolayfilt). APs were detected as peaks of amplitude more than 10 times the Quiroga threshold (QT), the median(|V(t)|/0.6745).

To develop a numerical routine, a group of human annotators identified 48 'high-quality' electrophysiology recordings. We then compiled a large set of descriptive statistics, listed below, and calculated the distribution of each of these statistics in the reference dataset, thereby defining an acceptable range expected of high-quality recordings. Each descriptive statistic was subsequently computed for recordings from all 213 neurons. Each recording was passed for further analysis if for all metrics it fell within the range spanned by the manually selected dataset of 48 recordings.

For each electrophysiology recording, we calculated 35 descriptive statistics.

Metrics computed on continuous electrophysiological data:

(1) Median relative deviation of the membrane potential (MRDM), the ratio between the median absolute deviation (MAD) and the median: MRDM = MAD(Vm)/median(Vm).

(2) Mean of the baseline (BL).

(3) Coefficient of variation of the baseline: std (BL)/mean (BL).

(4) Mean of the baseline noise, approximated by the QT (*Jewell et al., 2020*).

(5) Stability of the QT: thousand 10 s intervals were uniformly sampled from each recording, and the QT was computed on each sample. Quiroga noise stability (QNS) was defined as the coefficient of variation over the 1000 QT samples.

(6) $r^2$ of linear regression (MATLAB regression function) of the 1000 QT samples against the start times of the 10 s segments on which the QT was computed.

(7) Slope of linear regression (MATLAB regression function) of the 1000 QT samples against the start times of the 10 s segments on which the QT was computed.

(8) $r^2$ of linear regression (MATLAB regression function) of the baseline against time.

(9) Slope of linear regression (MATLAB regression function) of the baseline against time.

(10) The number of samples for which the baseline-subtracted trace exceeds the QT divided by the number of samples for which it dips below the negative of the QT.

Metrics computed on the AP time series. Only recordings with >3 APs were included:

(11) Number of APs.

(12) Maximum likelihood inter-AP interval (MATLAB lognfit function).

(13) Mean AP amplitude.

(14) AP amplitude coefficient of variation.

(15) AP amplitude median relative deviation.

(16) Relative AP amplitude range: (max[amplitude] – min[amplitude])/median(amplitude).

(17) AP amplitude max/min ratio: max(amplitude)/min(amplitude).

(18) Signal-to-noise ratio (SNR), median(amplitude)/QT.

Metrics computed on 2-ms-long AP waveforms, AP time ±1 ms smoothed with MATLAB smooth function with sgolay option:

(19) 'Left' width-half-max (LWHM), the mean width at half the amplitude before the detected AP time.

(20) 'Right' width-half-max (RWHM), the mean width at half the amplitude after the detected AP time.

(21) Full width at half amplitude (FWHM). FWHM = LWHM + RWHM.

(22) Coefficient of variation of LWHM.

(23) Coefficient of variation of RWHM.

(24) Coefficient of variation of FWHM.

(25) $r^2$ of linear regression (MATLAB regression function) of AP amplitude against AP time.

(26) Slope of linear regression (MATLAB regression function) of AP amplitude against AP time.

(27) $r^2$ of linear regression (MATLAB regression function) of AP FWHM against AP time.

(28) Slope of linear regression (MATLAB regression function) of AP FWHM against AP time.

Firing rate-based metrics. Firing rate was estimated by convolution of the AP train with a 1-s-long box-car window (MATLAB conv function):

(29) Mean firing rate (FR).

(30) Coefficient of variation of FR.

(31) $r^2$ of linear regression (MATLAB regression function) of firing rate against time.

(32) Slope of linear regression (MATLAB regression function) of firing rate against time.

(33) Pearson correlation (MATLAB corrcoef function) of BL vs. FR.

(34) Pearson correlation (MATLAB corrcoef function) between the baseline at AP time points and AP amplitude.

(35) Pearson correlation (MATLAB corrcoef function) between the baseline at AP times and the AP FWHM.

## Neuropil subtraction, high-resolution images

To approximate somatic fluorescence ($F_{cell\_true}$) without neuropil contamination, a scale version of the neuropil fluorescence ($F_{neuropil}$) was subtracted from each somatic fluorescence trace, after (*Akerboom et al., 2012*): $F_{cell\_true}(t) = F_{cell\_measured}(t) - r * F_{neuropil}(t)$. We determined the optimal scale factor (r) for neurons with GCaMP6f to be 0.82 (see 'Results' section). We therefore used r = 0.8 as our default scale factor. For some neurons, $F_{neuropil}$ was large enough relative to $F_{cell\_measured}$ that r = 0.8 resulted in negative fluorescence. For these neurons, we set r to 0.7, 0.6, or 0.5. For our dataset of 48 GCaMP6s and GCaMP6f neurons, we set r to 0.8 for 40 neurons, to 0.7 for four neurons, to 0.6 for three neurons, and to 0.5 for one neuron.

## Neuropil subtraction, downsampled images

Neuropil subtraction was performed as described for the Allen Brain Observatory (*de Vries et al., 2020*).

## Trace analysis

Electrophysiology and calcium imaging data were analyzed using custom MATLAB and Python scripts. For electrophysiology, Vm was filtered between 250 Hz and 5 kHz, and automated AP detection was performed using a threshold criterion (5×std of Vm).

For calcium imaging, in-plane motion artifacts were corrected (*Dombeck et al., 2007*), and neuron/ROI selection was performed using a semi-automatic algorithm (*Chen et al., 2013*) (kindly provided by Karel Svoboda, Janelia Research Campus). Ring-shaped ROIs were used to select GCaMP6-positive excitatory neurons, with GCaMP6 expression typically excluded from the nucleus and restricted to the cytoplasm.

To construct AP-calcium fluorescence response curves, we first identified all isolated AP events. For GCaMP6s, isolated events were separated from previous and subsequent events by $\geq$1000 and $\geq$500 ms, respectively. For GCaMP6f, isolated events were separated from previous and subsequent events by $\geq$300 ms. One result of finding isolated events is that only a minority of APs were used to construct AP-calcium fluorescence response curves. Within each event, APs were summed over 250 ms. Fluorescence traces were aligned to the first AP in each event, with t = 0 preceding the AP by <1 frame (6.3 ms at 158 Hz). For each event, $\Delta F/F = (F-F_{0,local})/F_{0,global}$, where $F_{0,local}$ was the mean fluorescence over 100 ms before the first AP, and $F_{0,global}$ was the minimum $F_{0,local}$ across trials. For GCaMP6s and GCaMP6f, peak $\Delta F/F$ was calculated by first finding $t_{max}$, the time of the maximum $\Delta F/F \leq$ 500 ms and 300 ms after the first AP, respectively. Peak $\Delta F/F$ was the mean $\Delta F/F$ from $t_{max}$ - 50 ms to $t_{max}$ + 50 ms. Bursts of >5 APs were excluded from analysis due to the low frequency of such events.

## Fluorescence-to-photon conversion

Mean and variance of the fluorescence, calculated pixelwise for each image, were linearly related, consistent with shot noise-limited imaging.

The resulting slope and offset of the least squares fit were used to convert fluorescence to number of photons: photons = (F − [−offset/slope])/slope (http://github.com/AllenInstitute/QC_2P). To

account for different pixel dwell times along the resonant scanning axis, photon gain and offset were computed pixel-by-pixel along the resonant axis.

## Trial-to-trial variability

For each neuron, fluorescence was summed over all somatic pixels and converted to photons. For each 1 AP event, mean photon count 0.1–0 s before the AP was subtracted. $t_{max}$, the time of the maximum photon count, was calculated from the mean 1 AP trace. Photon count in each trial was determined at $t_{max}$, and the 95% confidence interval was calculated as mean (across trials)$\pm 1.96 * \sqrt{\mathrm{meanpeak}}$. The percentage of trials with peak fluorescence outside the 95% confidence interval was used as a measure of trial-to-trial variability.

## Ground truth-optimized event detection

We compared fluorescence traces of the response (1 AP or 2 AP events) to that of 0 AP events (*Chen et al., 2013*). For each recording, the mean response trace was used as the template vector. The template vector was normalized after subtracting the mean to create the unit vector, and the scalar results of projecting the response and noise traces on the unit vector were computed: $r_i$ and $n_i$ for response and noise scalars, respectively. The detection threshold was defined as the xth percentile of $n_i$ values, where 1–x represented the false positive probability (e.g. x = 95 for 95th percentile or 5% false positive probability), and the detection probability (true positive probability) was the fraction of $r_i$ values above the detection threshold.

## Image downsampling

Fluorescence movies were sub-sampled by a factor of 4 in space and 5 in time (to pixel size 0.80 µm and frame rate 30.3 Hz) to match the sampling rate and approximate number of pixels per soma of the Allen Brain Observatory (*de Vries et al., 2020*). To assess the effect of downsampling on subsequent processing, 20 different approaches were tried in parallel (downsampling starting with the 1st, 2nd,..., 4th pixel × 1st, 2nd,..., 5th frame, respectively). 4 × 5 internally identical blocks, one block for each downsampling strategy, were tiled for a total of 400 almost identical ROIs per recording. Segmentation to find somatic ROIs, demixing of traces from nearby somata, neuropil subtraction, and the calculation of ΔF/F were performed as described for the Allen Brain Observatory (*de Vries et al., 2020*).

In cases where only the neuron of interest was found during segmentation, trace extraction would yield a family of 400 self-similar traces and the median was used for subsequent analysis. To catch cases where segmentation yielded additional objects that were not part of the neuron of interest, additional QC steps were required. The traces were first clustered using DBSCAN (*Ester et al., 1996*; *Schubert et al., 2017*), and each cluster median was compared against white noise of the same mean and standard deviation (KS test) and rejected as artifact if it was not significantly different (p<0.05). In cases where multiple clusters were significantly different from noise, this was either due to multiple neurons being present in the field of view or due to residual motion artifacts resulting in multiple translated copies of the same neuron. To disambiguate these two possibilities, the top three clusters were merged: sums were computed for all six possible combinations (sampled without replacement) of (up to) three most distinct cluster medians, and the combination most significantly correlated with the measured electrophysiological AP train was selected for subsequent analysis. Correlation significance was determined by building a null distribution of correlations between the cluster medians and 1000 random Poisson trains with a rate matching that of the recorded AP train. If there was no more significant correlation between any cluster median (or sum thereof) and the measured AP train than the 0.5th percentile of the null distribution (i.e. p>0.005), the recording was failed. Finally, we eliminated from further analysis <10 neurons with an abrupt and sustained (seconds) rise in spike rate and subsequent loss spiking activity out of concern that this activity pattern might indicate a breached plasma membrane.

To compare the noise characteristics of the downsampled images to the Allen Brain Observatory, we computed the robust standard deviation, a median-based method with outlier removal (*de Vries et al., 2020*). For the Allen Brain Observatory, we analyzed fluorescence over periods in which there were no apparent AP-evoked changes in fluorescence.

## Acknowledgements

We are grateful for the Animal Care, Transgenic Colony Management, and Lab Animal Services teams for mouse husbandry, and Carol Thompson and John Phillips for providing project management support. We thank Karel Svoboda, Hod Dana, and Tsai-Wen Chen for sharing analysis software. This work was funded by Allen Institute for Brain Science. This work was also supported by grants from National Institutes of Health (R01EB026908) to MAB, and from National Natural Science Foundation of China (NSFC31871055) and Guangdong Science and Technology Department (2020B1212060018, 2017B030314026 and 2018B030334001) to LL. We thank the Allen Institute founders, Paul G Allen and Jody Allen for their vision, encouragement, and support.

## Additional information

### Funding

| Funder | Grant reference number | Author |
|---|---|---|
| Allen Institute for Brain Science | Program funds | Lawrence Huang<br>Peter Ledochowitsch<br>Ulf Knoblich<br>Jérôme Lecoq<br>Gabe J Murphy<br>R Clay Reid<br>Saskia EJ de Vries<br>Christof Koch<br>Hongkui Zeng<br>Michael A Buice<br>Jack Waters<br>Lu Li |
| National Natural Science Foundation of China | NSFC31871055 | Lu Li |
| Guangdong Science and Technology Department | 2017B030314026 | Lu Li |
| Guangdong Science and Technology Department | 2018B030334001 | Lu Li |
| Guangdong Science and Technology Department | 2020B1212060018 | Lu Li |
| National Institutes of Health | R01EB026908 | Michael A Buice |

This work is funded by the Allen Institute for Brain Science. The funder had no role in study design, data collection and interpretation, or the decision to submit the work for publication.

### Author contributions

Lawrence Huang, Peter Ledochowitsch, Data curation, Formal analysis, Investigation, Visualization, Methodology, Writing - original draft; Ulf Knoblich, Data curation, Investigation, Methodology; Jérôme Lecoq, Formal analysis, Investigation, Writing - review and editing; Gabe J Murphy, Formal analysis, Supervision; R Clay Reid, Conceptualization, Supervision; Saskia EJ de Vries, Formal analysis; Christof Koch, Conceptualization, Supervision, Funding acquisition, Writing - review and editing; Hongkui Zeng, Conceptualization, Formal analysis, Supervision, Investigation, Writing - review and editing; Michael A Buice, Formal analysis, Supervision, Investigation, Methodology, Writing - review and editing; Jack Waters, Formal analysis, Investigation, Methodology, Writing - review and editing; Lu Li, Data curation, Formal analysis, Supervision, Investigation, Methodology, Writing - review and editing

### Author ORCIDs

Peter Ledochowitsch (iD) https://orcid.org/0000-0003-0835-3444
R Clay Reid (iD) https://orcid.org/0000-0002-8697-6797
Saskia EJ de Vries (iD) https://orcid.org/0000-0002-3704-3499

Hongkui Zeng  https://orcid.org/0000-0002-0326-5878

Michael A Buice  https://orcid.org/0000-0002-2196-1498

Jack Waters  https://orcid.org/0000-0002-2312-4183

### Ethics

Animal experimentation: Experimental procedures were conducted in accordance with NIH guidelines and approved by the Institutional Animal Care and Use Committee (IACUC) of the Allen Institute for Brain Science under protocol number 1509.

### Decision letter and Author response

Decision letter https://doi.org/10.7554/eLife.51675.sa1

Author response https://doi.org/10.7554/eLife.51675.sa2

## Additional files

### Supplementary files

• Transparent reporting form

### Data availability

All data generated and analyzed in this study are available at https://portal.brain-map.org/explore/circuits/oephys.

The following dataset was generated:

| Author(s) | Year | Dataset title | Dataset URL | Database and Identifier |
|---|---|---|---|---|
| Huang L, Ledochowitsch P, Knoblich U, Lecoq Jrm, Murphy GJ, Reid RC, de Vries SEJ, Koch C, Zeng H, Buice MA, Waters J, Li L | 2019 | Ophys/Ephys Calibration Data | https://portal.brain-map.org/explore/circuits/oephys | Allen Brain Map, circuits/oephys |

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
