## [Decision Letter]

**Acceptance summary:**

The study by Huang, Knoblich et al. represents an important contribution to the field, providing critical examination of in vivo 2-photon calcium imaging for the detection of underlying spike events. Overall, the work is very high quality. The demonstration that spike detection is ~15% under normal "low zoom" imaging conditions is a stunning observation that should be a wake-up call to large parts of the community. The results are somewhat sobering for investigators in the sense that no once-size-fits all strategy accurately extracted spiking in commonly-used conditions from fluorescence data.

**Decision letter after peer review:**

Thank you for submitting your article "Relationship between spiking activity and simultaneous fluorescence signals in transgenic mice expressing GCaMP6" for consideration by *eLife*. Your article has been reviewed by three peer reviewers, and the evaluation has been overseen by Gary Westbrook as the Senior Editor. The following individuals involved in review of your submission have agreed to reveal their identity: Karel Svoboda (Reviewer #1); Michael Higley (Reviewer #2); Bernardo L Sabatini (Reviewer #3).

The reviewers have discussed the reviews with one another and the Senior Editor has drafted this decision to help you prepare a revised submission.

Summary

This manuscript explores an important topic with data that is difficult to obtain. All reviewers thought the manuscript was worth publishing in *eLife* after appropriate revisions. However all reviewers had concerns (some overlapping) that are important to address. Many of comments can be addressed with clarifications, rewording and better explanation/analysis of some aspects of the work. However, the manuscript contains some statements and conclusions that indicate incomplete understanding of sources of noise in the imaging experiments. This is important because the paper is really focused on detection. The full comments of the reviewers are below.

Reviewer #1:

Calcium imaging is widely used to track activity in large populations of neurons. Calcium-dependent fluorescence is often thought of as “activity”, but it is unclear what this means because the spike to fluorescence transform is not well understood. As a result the interpretation of calcium imaging data is often superficial and misleading. This is in part because ground truth data (i.e. simultaneous imaging and recording) is scarce. The major contribution of this paper is the report of a substantial bolus of additional ground truth data in several widely used transgenic mouse lines. This is hard-won data and I support publication. But some work is required first.

The take-home messages in this paper are: There are differences in spike detection across mouse lines, with 6s-expressing mice outperforming 6f-expressing mice. This is expected.

There are differences in detection across mice expressing the same indicator:

These differences across mice expressing the same indicator are not explained. My suspicion is that differences in neuropil (i.e. background) is likely the culprit (emx1 expresses in L4 – with lots of axons in L2/3; CamK2 does not). Explain this and the strange claim that the noise level is different in the emx1-s vs the tetO-s mice.

There is variability in the response to single spikes:

No attempt is made to distinguish interesting biological sources of noise (e.g. spike calcium coupling) to non-interesting biology (movement) and non-biological explanations (instrumentation). The paper would be stronger if the sources of noise were analyzed better. In particular, are the measurements done in a shot-noise limited regime? Does movement contribute? How about other instrumentation noise (shouldn't but still)?

Lower zoom imaging produces lower snr than higher zoom imaging

The section “Comparing spike-to-calcium fluorescence response curves imaged at high and low spatiotemporal resolutions” is also strange. It's not clear to me what exactly the point is here.

Obviously, everything else being equal, increasing the fov from 20 – 400 μm reduces the light dose per neuron by a factor of 400 and thus the SNR by a factor of 20 (see Peron et al., CONB 2015) just based on shot-noise alone! No one would image a 20 μm FOV with the same power as a 400 μm FOV.

Reviewer #2:

The study by Huang, Knoblich et al. represents a very important contribution to the field, providing critical examination of in vivo 2-photon calcium imaging for the detection of underlying spike events. Overall, the work is very high quality, and I have no concerns or suggestions with regard to data collection. I do have several major points on analyses that need to be addressed, detailed below. Overall, the work reads as if the major focus is the comparison of different transgenic GCaMP6 lines. While this topic is interesting, the far more important issue is the ability to estimate spiking from imaging data under "real world" conditions. Thus, far more emphasis needs to be placed on the "low zoom" data. The difference between the mouse lines is modest at best. However, the demonstration that spike detection is ~15% under normal imaging conditions is a stunning observation that should be a wake-up call to large parts of the community.

1) As noted, the high impact value of the study is on the "low zoom" data, as this represents the situation for the vast majority of experimental labs using GCaMP6. All analyses in the manuscript, including the examples comparing ΔF/F and spiking (Figures 1-6) need to be repeated for the low-zoom data. The analysis of neuropil correction is absolutely critical, as this may play a much larger role in the reduced spatiotemporal sampling regime. I would actually suggest making these analyses the major focus, rather than limited to Figure 7.

2) The paired statistical comparisons of single spike signals with a random period (e.g., Figures 4—figure supplement 1 and Figure 7—figure supplement 1) are not very informative. The fact that there is an average difference is far less important than the discriminability of true spikes from noise.

3) It is unclear how the 91 selected cells were chosen for "high quality recording and imaging". It would be useful to know how the results change for "lower quality" imaging, as this may better inform experimentalists on data collection.

4) The authors should make some attempt to explain why the spike detection is so much poorer at low zoom. Is it the fewer pixels per cell (factor of 16) or the lower sampling rate (factor of 4-5). Disambiguating these contributors would better help the field. For example, how does spatially or temporally down-sampling the high-zoom data affect spike detection?

5) The sensitivity and ROC analyses assume that the only way to extract spike information from a fluorescence trace is to do a linear thresholding on ΔF/F amplitude, but many spike extraction methods take into account multiple properties of the shape of fluorescence transients. It would be beneficial if the authors could apply some of the most commonly used spike extraction algorithms to their data in order to benchmark/validate them (particularly under the low-zoom conditions).

6) Please address the accuracy of spike detection under low-zoom for all locations across the FOV. At low zoom, most commercial two-photon microscopes have increased PSF and reduced photon collection efficiency at the edges of a large field.

7) Please explain why the imaging data were temporally smoothed. This is non-standard, and it is important to know how the analyses apply to conventional approaches.

8) It is unclear why the tetO-GCaMP6s have a higher noise floor. Are there any systematic differences in the way these data were collected (different anesthetic, different depth, different amounts of brain motion, different age of mice) that could explain this?

9) Please state the ages of each individual animal used in this study. Is age a confound? Are the distribution of ages for each mouse line matched? For viral expression of GCaMP, time of expression is an important variable. What role does it play in transgenic mice?

Reviewer #3:

This is a well done and systematic comparison of the relationship between electrophysiologically recorded action potentials and GECI-reported fluorescence transients in a variety of transgenic mouse lines used for such recordings. The authors carefully compare the ability of detect single and 5 spike events. Many people will read this study and find its results useful in designing their own experiments and in analyzing their data.

A few points need consideration

1) The authors only use 91 out of the of the 237 cells collected. If there was selection made based on the quality of the imaging data, this may strongly impact the results. How were the cells chosen for inclusion? What happens if the other cells are analyzed?

2) Only peri-cell annular neuropil fluorescence correction is attempted. What if CNMF-type algorithms are used? Does this yield substantially different results? It is not clear that the fixed r-value subtractive approach is necessarily the best when trying to detect single spikes. (I am happy to be convinced otherwise if I am wrong).

[Editors' note: further revisions were suggested prior to acceptance, as described below.]

Thank you for resubmitting your article "Relationship between simultaneously recorded spiking activity and fluorescence signal in GCaMP6 transgenic mice" for consideration by *eLife*. Your revised article has been reviewed by three peer reviewers, and the evaluation has been overseen by Gary Westbrook as the Senior Editor and Reviewing Editor. The following individuals involved in review of your submission have agreed to reveal their identity: Karel Svoboda (Reviewer #1); Michael Higley (Reviewer #2); Bernardo L Sabatini (Reviewer #3). The reviewers have discussed the reviews with one another and the Reviewing Editor has drafted this decision to help you prepare a revised submission.

We would like to draw your attention to changes in our revision policy that we have made in response to COVID-19 (https://elifesciences.org/articles/57162). First, because many researchers have temporarily lost access to the labs, we will give authors as much time as they need to submit revised manuscripts. We are also offering, if you choose, to post the manuscript to bioRxiv (if it is not already there) along with this decision letter and a formal designation that the manuscript is "in revision at *eLife*". Please let us know if you would like to pursue this option.

Summary

The reviewers all agreed that this data is an important resource. This manuscript reports a valuable simultaneous ephys-ophys dataset with a great number of cells recorded (N = 237) and selected (N = 91). This data set will support the development of more refined spike-to-fluorescence or fluorescence-to-spike inference models. However, inclusion of data collected under more "real world" conditions – meaning low zoom and typical laser power – would substantially increase the impact. As presented, the results remain somewhat limited without such data for comparison. The authors were requested to examine how different spike detection and neuropil subtraction algorithms would change their conclusions. Instead of including that analysis here, they have instead posted another manuscript to BioRxiv. Although this makes the analysis public, it fails to improve this manuscript.

A number of specific comments below require your attention before we can make a final decision regarding publication in *eLife*.

Essential revisions

1) There is really only minimal analysis of the data. It would be greatly improve the impact of this work to compare detailed spike to fluorescence model parameters to other measurements using the models in Wei et al., 2019.

2) From this and previous studies it is clear that viral expression of GCamp provides better SNR than transgenic expression. This remains a mystery and would be worthy of comment in the manuscript

3) Of course, it is expected that SNR will decrease with zoom at constant power because of shot noise. But it is highly unlikely that investigators would use the same laser power for imaging over changes in zoom by a factor of 20. Please comment.

4) Critical references are missing. Dana et al., 2014 and Wei et al., 2019 have shown that “GCaMP6s cells have spike-triggered fluorescence responses of larger amplitude, lower variability and greater single-spike detectability than GCaMP6f” in transgenic mice.

5) Important spike-to-calcium parameters (e.g. rise- and decay- time constants) are missing. These parameters should be reported as part of the basic analyses; please incorporate the analyses e.g. Figures 2D,E,F and 3F in Chen et al., 2013. One can also use existing spike-to-fluorescence models to estimate the parameters (Wei et al., 2019).

6) Spike-event snippet creation. First, the authors should demonstrate how they chose the parameters for the snippets in each imaging condition. It is not clear that the choice of snippet parameters were optimal; for example, Emx1-s and tetO-s ΔF/F at 4APs in Figure 2 seem not to reach the maximum within 200 ms window. Ideally, these parameters could be determined by the estimates of rise- and decay- time constants. For example, if the half-rise time is 100 ms and the spike-event snippet is 200 ms, one should use a time window at least 400 ms to capture the peak dF/F.

7) Neuropil removal. This is a confusing part throughout the manuscript. At the beginning, the authors preferred not performing neuropil correction because it might increase peak ΔF/F variability (Figure 2—figure supplement 1D), then in the later part, the authors claimed that the importance of the neuropil correction to the spike detection. The authors could offer a systematic study to address if neuropil should be removed and how. It seems like the optimal r for each cell should be used throughout the manuscript. One study from Kerlin et al., 2010 (this paper should be cited) provided some answers to this.

8) ΔF/F computation. In general one would take a long time-window to compute F0 with the background subtraction in the denominator, where F0 aims to reflect 0AP fluorescence. Computing locally as the mean within 50 or 20 ms before the first spike event is risky, because F0 in a short window can be contaminated by the previous calcium decay after a many-AP event e.g. F0 around 5-AP in Figure 1B.

9) Peak ΔF/F variability. First, it is not clear how the mean coefficient of variation of ΔF/F peak (a main measure of peak ΔF/F variability throughout the manuscript) was defined and computed. Is the coefficient in the term of the number of spikes? How was it computed in a given nAP case, e.g. Figure 4A right or Figure 4B? Second, although shot noise is dominant on single pixels, or at high sample rates or in low brightness conditions, variability would be reduced when computing F by averaging over N pixels. The analysis on single pixels is somewhat misleading. Third, trial-by-trial variability is important but the measurement is not clear. In general, the peak ΔF/F should depend on the time series of the spikes. It roughly depends on the number of the spikes in a brief time window. One would thus expect two sources of trial-by-trial variance, one depends on the number of spikes; the other depends on the spike pattern as the number of spikes is fixed. Authors should be able to decompose the variability into these terms.

10) The finding that variance in response amplitude is no larger than expected from shot-noise is surprising and unlikely to be true. There are many reasons why the coupling between AP and Ca might be variable (modulation, baseline potential, state of channels, channel fluctuations). Make sure this analysis is correct.

11) Lastly, as demonstrated above, the peak ΔF/F does not directly depend on behavioral condition when the spike pattern is given. Yet the difference in spon vs visstim in Cux2-f mice is striking. Can this difference be explained by the spike pattern difference in spon vs visstim conditions? If not, what causes this difference?

[Editors' note: further revisions were suggested prior to acceptance, as described below.]

Thank you for resubmitting your work entitled "Relationship between simultaneously recorded spiking activity and fluorescence signal in GCaMP6 transgenic mice" for further consideration by *eLife*. Your revised article has been evaluated by Gary Westbrook (Senior Editor) and Reviewers 2 and 3 from the original submission.

We are satisfied with the data in the revised manuscript, but ask that you revise the text in accord with the comments below as we believe that these changes with substantially improve the impact of your important work.

Reviewer #2:

Far more emphasis needs to be placed on the "low zoom" data. The difference between the mouse lines is modest at best. However, the demonstration that spike detection is ~15% under normal imaging conditions is a stunning observation that should be a wake-up call to large parts of the community.

Reviewer #3:

The authors have done a great deal to improve the study. There are some remaining points can be easily addressed to improve the study even further.

1) There is no description of what cells the Cux2 and Emx1 cre lines target. It would help make the study more approachable and useful to include this information here.

2) The authors report that there is no epileptiform activity in the mice they use. Do they not see such activity in these mice in their hands or do they select individual mice without seizures? How to they judge the presence of epileptiform activity?

3) In light of point 2, the fraction of spikes that are in groups of 1-5 spike bursts is very different for GCAMP6s and GCAMP6f mice. This is a metric measured by the cell-attached recording. Doesn't this indicate that the transgene has a large effect on cellular or circuit activity patterns?

4) The authors use QC metrics to reduce the number of cells analyzed from >200 to 48. Can they report what factors result in 75% of the cells being rejected? Do they know which rejection factors actually impact the ability to accurately infer spiking from fluorescence? Such insight would be very useful for others who don't have cell-attached recordings but want to be able to understand what cells to include in their final analyses.

5) Lastly, the Discussion leaves something to be desired. It is a synopsis of the conclusions and iterates some rather obvious points but also making some claims without backing. It is not clear that the statements about GFP quantum yield being the limiting factor to further improvement of single-spike detection is correct. The Ca-sensitive fluorophores may not get brighter, but their linearity, DF/F, stability etc may improve. Given the assumption that photon-budget is limiting, the statement that voltage-sensors may provide the solution seems wrong as these will provide many fewer photons per AP. The authors could use the discussion instead of provide some helpful hints to imagers to make the best use of their data based on the analyses presented.

---

## [Author Response]

SummaryThis manuscript explores an important topic with data that is difficult to obtain. All reviewers thought the manuscript was worth publishing in eLife after appropriate revisions. However all reviewers had concerns (some overlapping) that are important to address. Many of comments can be addressed with clarifications, rewording and better explanation/analysis of some aspects of the work. However, the manuscript contains some statements and conclusions that indicate incomplete understanding of sources of noise in the imaging experiments. This is important because the paper is really focused on detection. The full comments of the reviewers are below.

We thank the reviewers for their constructive comments and suggestions, which have been extremely helpful in our effort to improve the manuscript. In the revised manuscript submitted here, we have attempted to address all the points raised by the reviewers. In particular, the major part of newly added analyses is focused on sources of noise in high and low zoom imaging experiments. Here we provide a brief summary of the major changes we have made:

Removed insufficiently substantiated conclusions about the difference between tetO-s and other mouse lines (sample size too small for tetO-s) and about the difference in response curve slope between high zoom and low zoom data (optimal neuropil subtraction may be different between zooms and was not considered). Also removed several less informative results to streamline the paper.

Added new measurements and analyses to demonstrate how signal detection difference between high and low zoom imaging conditions can indeed be explained by shot noise calculations for most cells.

Referred the reviewers to our follow-up manuscript, now available on bioRxiv, which includes systematic subsampling of our high zoom data to model the typical population calcium imaging data and application of state-of-the-art spike inference methods on the resampled data.

Clarified our quality control process that resulted in the selection of the 91 cells from our larger dataset for further analysis.

Reviewer #1:Calcium imaging is widely used to track activity in large populations of neurons. Calcium-dependent fluorescence is often thought of as “activity”, but it is unclear what this means because the spike to fluorescence transform is not well understood. As a result the interpretation of calcium imaging data is often superficial and misleading. This is in part because ground truth data (i.e. simultaneous imaging and recording) is scarce. The major contribution of this paper is the report of a substantial bolus of additional ground truth data in several widely used transgenic mouse lines. This is hard-won data and I support publication. But some work is required first.The take-home messages in this paper are: There are differences in spike detection across mouse lines, with 6s-expressing mice outperforming 6f-expressing mice.This is expected.There are differences in detection across mice expressing the same indicator:These differences across mice expressing the same indicator are not explained. My suspicion is that differences in neuropil (i.e. background) is likely the culprit (emx1 expresses in L4 – with lots of axons in L2/3; CamK2 does not). Explain this and the strange claim that the noise level is different in the emx1-s vs the tetO-s mice.

Some of our analyses were from small numbers of recordings. The differences in spike detection and noise level across mice are examples, with the tetO-s results based on 4 cells from 1 mouse. Differences between mouse lines were statistically significant, as reported in the original manuscript, but the small number of recordings leaves us with little confidence in the apparent differences in spike detection and noise level in the Emx1-s vs tetO-s mice. Thus, we have removed these conclusions from the revised manuscript.

We think differences between neuropil fluorescence is unlikely to be due to the difference between Emx1 and Camk2a. GCaMP6s expression in Emx1-s (*Emx1-IRES-Cre;Camk2a-tTA;Ai94*) and tetO-s (*Camk2a-tTA;tetO-GCaMP6s*) lines are both under control of the Camk2a promoter, since Emx1-Cre has broader expression in cortex than Camk2a-tTA (as the reviewer mentioned) and thus Camk2a is the same restricting factor in both lines.

There is variability in the response to single spikes:No attempt is made to distinguish interesting biological sources of noise (e.g. spike calcium coupling) to non-interesting biology (movement) and non-biological explanations (instrumentation). The paper would be stronger if the sources of noise were analyzed better. In particular, are the measurements done in a shot-noise limited regime? Does movement contribute? How about other instrumentation noise (shouldn't but still)?

We have expanded our noise analyses and present them in the new Figure 4—figure supplement 1.

Our new analyses indicate that shot noise is the dominant noise source in most cells (Figure 4—figure supplement 1). Furthermore, most of the trial-to-trial variance in the amplitudes of calcium transients can be attributable to shot noise. Trial-to-trial variance was 0-2% greater than expected from shot noise (variance was greater than expected by 1±14% in Emx1-s, 0±8% in tetO-s and 2±7% in Cux2-f). This 0-2% additional variance is presumably the summed effects of motion, instrumentation noise and spike-to-calcium coupling. Even without separating these three minor contributions, we can conclude that there’s little trial-to-trial variability in spike-to-calcium coupling in 3 of our 4 mouse lines.

The exception is Emx1-f, in which the trial-to-trial variance is 48±91% greater than expected from shot noise. We believe the increased variability in Emx1-f may be biological (and possibly related to this line’s susceptibility to epileptiform activity) and we explore this topic further in our follow-up paper, Ledochowitsch et al. (available on bioRxiv; https://www.biorxiv.org/content/10.1101/800102v1).

Lower zoom imaging produces lower snr than higher zoom imagingThe section “Comparing spike-to-calcium fluorescence response curves imaged at high and low spatiotemporal resolutions” is also strange. It's not clear to me what exactly the point is here.Obviously, everything else being equal, increasing the fov from 20 – 400 μm reduces the light dose per neuron by a factor of 400 and thus the SNR by a factor of 20 (see Peron et al. CONB 2015) just based on shot-noise alone! No one would image a 20 μm FOV with the same power as a 400 μm FOV.

In this study, we conducted both high and low zoom imaging for a subset of the cells. When switching zooms we did NOT change the laser power. As a result, the laser power used for our low zoom data may be lower than that of typical population-scale imaging experiments. Indeed, our low zoom data had lower median photon flux compared to the same mouse lines in the Allen Brain Observatory dataset. We have kept the low zoom data in the manuscript, for reasons explained below in response to this reviewer’s further comments, as well as to address some of reviewer #2’s comments on this topic.

We have now added the following analysis in the low zoom Results section. Our expected reduction of SNR between high zoom and low zoom is by a factor lower than sqrt(400) = 20. After accounting for the duty cycle ratio for photon collection between the two zoom conditions (high/low = 0.7) and for the ratio of ROI areas drawn separately for the two zoom conditions, we find the effective ratio of photon fluxes (high/low) to be ~333. Considering further that the ratio of sampling rates for the fluorescence traces is 158/30 = 5.3, the expected SNR ratio for the fluorescence traces (high/low) is actually sqrt(333 / 5.3) = 7.9. Most cells for which we have both high and low zoom recordings, closely match that expected SNR ratio. There are a few outlier cells where we suspect that z-drift and/or activity differences between the high and the low zoom measurements lead to deviating SNR ratios.

Reviewer #2:The study by Huang, Knoblich et al. represents a very important contribution to the field, providing critical examination of in vivo 2-photon calcium imaging for the detection of underlying spike events. Overall, the work is very high quality, and I have no concerns or suggestions with regard to data collection. I do have several major points on analyses that need to be addressed, detailed below. Overall, the work reads as if the major focus is the comparison of different transgenic GCaMP6 lines. While this topic is interesting, the far more important issue is the ability to estimate spiking from imaging data under "real world" conditions. Thus, far more emphasis needs to be placed on the "low zoom" data. The difference between the mouse lines is modest at best. However, the demonstration that spike detection is ~15% under normal imaging conditions is a stunning observation that should be a wake-up call to large parts of the community.1) As noted, the high impact value of the study is on the "low zoom" data, as this represents the situation for the vast majority of experimental labs using GCaMP6. All analyses in the manuscript, including the examples comparing ΔF/F and spiking (Figures 1-6) need to be repeated for the low-zoom data. The analysis of neuropil correction is absolutely critical, as this may play a much larger role in the reduced spatiotemporal sampling regime. I would actually suggest making these analyses the major focus, rather than limited to Figure 7.

The revised paper includes a more thorough analysis of the low zoom data in the final Results section, “Spike detection during population imaging, at low zoom”. During the analysis process we realized that our high and low zoom data were collected at the same laser power. As a result, the low zoom data have lower photon flux and poorer SNR than many population imaging studies, limiting the conclusions we can draw about spike detection under common experimental conditions. Thus, our basic characterization is still mainly on the high zoom data.

In this revision we further investigated if the reduction in single spike detection rates in our low zoom data were directly correlated with the reduction of photon flux and found this indeed to be the case, as the expected spike detection rates for low zoom through simulation of the high zoom data matched the measured spike detection rates from the low zoom data (Figure 6C). This result then allowed us to make a simple estimate of the spike detection rates under more common 2-photon imaging conditions which use higher laser power than in our low zoom experiment. By calculating per cell photon flux directly from the Allen Brain Observatory data, we were able to estimate the single spike detection rates to be 25-35% for GCaMP6f cells in that dataset (Figure 6D). We believe that this may be a more reasonable estimate of spike detection with GCaMP6 in many population imaging studies.

We agree that there’s a need to further address spiking estimates under “real world” conditions. The vast majority of our images were collected under high zoom conditions, with inherently higher SNR than most population imaging studies. To mimic common population imaging conditions, we have resampled our high-zoom recordings both spatially and temporally. The noise profile of this resampled data set matches that of the Allen Brain Observatory and thus approximates typical population-scale imaging conditions. Our study of resampled data has become sizable and includes a comparison of spike extraction algorithms (see point 5, below). Rather than merge this complex study with the current manuscript, we have written a second manuscript that focuses more directly on spike extraction from low-zoom images, the latter created by resampling high zoom images. The follow-up manuscript, Ledochowitsch et al., is available as a preprint on bioRxiv (https://www.biorxiv.org/content/10.1101/800102v1).

2) The paired statistical comparisons of single spike signals with a random period (e.g., Figures 4—figure supplement 1 and Figure 7—figure supplement 1) are not very informative. The fact that there is an average difference is far less important than the discriminability of true spikes from noise.

We agree with the reviewer and have removed these results and figure panels to streamline the manuscript.

3) It is unclear how the 91 selected cells were chosen for "high quality recording and imaging". It would be useful to know how the results change for "lower quality" imaging, as this may better inform experimentalists on data collection.

Cell selection was based on manual assessment. Selected cells exhibited (1) no motion artifacts (after motion correction in x-y axes); (2) no photobleaching; (3) no evidence of dye filling from the pipette; and (4) stable baseline and distinguishable spikes for electrophysiological recordings. We have added this description to the manuscript.

4) The authors should make some attempt to explain why the spike detection is so much poorer at low zoom. Is it the fewer pixels per cell (factor of 16) or the lower sampling rate (factor of 4-5). Disambiguating these contributors would better help the field. For example, how does spatially or temporally down-sampling the high-zoom data affect spike detection?

Poorer spike detection results from decreased SNR at low zoom. Since our images are shot noise limited (see responses to reviewer 1, and Figure 4—figure supplement 1), spatial and temporal downsampling are equivalent. (The effects of spatial and temporal downsampling might be different if one dimension were severely undersampled. For example, if temporal sampling were so slow that the peaks of some calcium transients were not sampled. We’ve not downsampled sufficiently in either space or time to enter this regime.) In switching from high to low zoom, spatial sampling was reduced more than temporal sampling (16x vs 5x) so the reduced photon flux at low zoom is primarily the result of reduced spatial sampling. sqrt(16) = 4 and sqrt(5) = 2.24 so spatial downsampling accounts for ~65% of the effect.

5) The sensitivity and ROC analyses assume that the only way to extract spike information from a fluorescence trace is to do a linear thresholding on ΔF/F amplitude, but many spike extraction methods take into account multiple properties of the shape of fluorescence transients. It would be beneficial if the authors could apply some of the most commonly used spike extraction algorithms to their data in order to benchmark/validate them (particularly under the low-zoom conditions).

We agree with the reviewer. The linear thresholding of ΔF/F amplitude is a basic form of spike extraction and applying more advanced spike inference algorithms might be informative. However, spike inference is a complex and active area of research. Our follow-up manuscript, Ledochowitsch *et al.*, includes a comparison of several spike extraction algorithms, using the ground-truth dataset we present here. We think that publishing a second manuscript, where we can address spike inference specifically, is preferable to squeezing a more sophisticated analysis of spike inference into the current manuscript.

6) Please address the accuracy of spike detection under low-zoom for all locations across the FOV. At low zoom, most commercial two-photon microscopes have increased PSF and reduced photon collection efficiency at the edges of a large field.

Unfortunately, we do not have data to address this point, as all our patched cells were near the center of the FOV.

7) Please explain why the imaging data were temporally smoothed. This is non-standard, and it is important to know how the analyses apply to conventional approaches.

Our frame rate under high zoom conditions was 158 frames per second (fps), approximately 5 times the ~30 fps rate of many modern 2-photon imaging experiments. We therefore used a smoothing window of 5 frames to approximate the temporal information more typical of 2-photon imaging experiments.

8) It is unclear why the tetO-GCaMP6s have a higher noise floor. Are there any systematic differences in the way these data were collected (different anesthetic, different depth, different amounts of brain motion, different age of mice) that could explain this?

There were no systematic differences in the way tetO-s data were collected, with no differences in mouse age, surgical preparation, or cell depths. As discussed above (in our responses to reviewer 1), these results suffered a small sample size and we have removed them from the manuscript.

9) Please state the ages of each individual animal used in this study. Is age a confound? Are the distribution of ages for each mouse line matched? For viral expression of GCaMP, time of expression is an important variable. What role does it play in transgenic mice?

We have added the ages of individual animals used in the study (Figure 1—figure supplement 1A). There were no significant differences in the age distributions across mouse lines. All mice were adults and one advantage of using transgenic mice is that transgene expression remains stable in adult mice over many months, as shown in our previous studies (Madisen et al., 2015; Daigle et al., 2018).

Reviewer #3:This is a well done and systematic comparison of the relationship between electrophysiologically recorded action potentials and GECI-reported fluorescence transients in a variety of transgenic mouse lines used for such recordings. The authors carefully compare the ability of detect single and 5 spike events. Many people will read this study and find its results useful in designing their own experiments and in analyzing their data.A few points need consideration1) The authors only use 91 out of the of the 237 cells collected. If there was selection made based on the quality of the imaging data, this may strongly impact the results. How were the cells chosen for inclusion? What happens if the other cells are analyzed?

Cell selection was based on manual assessment. Selected cells exhibited (1) no motion artifacts (after motion correction in x-y axes); (2) no photobleaching; (3) no evidence of dye filling from the pipette; and (4) stable baseline and distinguishable spikes for electrophysiological recordings. We have added this description to the manuscript.

2) Only peri-cell annular neuropil fluorescence correction is attempted. What if CNMF-type algorithms are used? Does this yield substantially different results? It is not clear that the fixed r-value subtractive approach is necessarily the best when trying to detect single spikes. (I am happy to be convinced otherwise if I am wrong).

We implemented CNMF (via CaImAn) to obtain fluorescence traces to compare to our fixed r-value subtractive approach. We have not found evidence that CNMF provides superior spike detection. However, we cannot be sure that our ROI segmentation and fluorescence trace extraction using CNMF was optimal. In fact, the segmentation was not robust to small changes in input parameters, leading us to suspect that trace extraction might be far from optimal.

Unfortunately, we do not have a reliable method of tuning CNMF parameters. There are many input parameters and we have found that there are complex interactions between them so finding optimal parameters would require an extensive search of segmentation across n-dimensional space, for each dataset. Since we do not know whether our CNMF results are optimal, we cannot, in principle, be confident that CNMF could not offer improved performance.

In conclusion, we have neither evidence that CNMF is superior for spike detection nor that it’s inferior and we decided, therefore, not to add CNMF results to the manuscript.

[Editors' note: further revisions were suggested prior to acceptance, as described below.]SummaryThe reviewers all agreed that this data is an important resource. This manuscript reports a valuable simultaneous ephys-ophys dataset with a great number of cells recorded (N = 237) and selected (N = 91). This data set will support the development of more refined spike-to-fluorescence or fluorescence-to-spike inference models. However, inclusion of data collected under more "real world" conditions – meaning low zoom and typical laser power – would substantially increase the impact. As presented, the results remain somewhat limited without such data for comparison. The authors were requested to examine how different spike detection and neuropil subtraction algorithms would change their conclusions. Instead of including that analysis here, they have instead posted another manuscript to BioRxiv. Although this makes the analysis public, it fails to improve this manuscript.

To address these concerns, we made three major changes to the manuscript:

1) We removed low zoom data from the manuscript. The laser power used for low zoom imaging was lower than typically used for population imaging, bringing into question the utility of the low zoom images. We replaced low zoom images with downsampled images (originally presented in our companion bioRxiv paper, Ledochowitsch *et al.*), as requested by reviewer 2 in the first round of reviews. We downsampled high zoom images to match the pixel size, frame rate and SNR of the Allen Brain Observatory and we believe these downsampled images are more representative of typical “real world” population imaging conditions.

2) We adopted the more rigorous quality control criteria outlined in Ledochowitsch *et al.*, resulting a smaller number of neurons in our updated dataset, now 48 neurons. More information on numbers of recordings, broken down by genotype, is provided in Table 1.

3) We include analysis of spike detection. We applied three leading spike inference models and compared event detection under ideal (high zoom) and real-world-like (downsampled) conditions.

Essential revisions1) There is really only minimal analysis of the data. It would be greatly improve the impact of this work to compare detailed spike to fluorescence model parameters to other measurements using the models in Wei et al., 2019.

We have added analyses, the major addition being a comparison of performance of three spike inference models on high zoom and downsampled datasets (Figures 6 and 7). Our main conclusion is that only ~20-30% of events are detected, at 1% false positive probability, under population imaging conditions.

2) From this and previous studies it is clear that viral expression of GCamp provides better SNR than transgenic expression. This remains a mystery and would be worthy of comment in the manuscript

In the revised manuscript we have reanalyzed our data with proper neuropil subtraction and time windows following the reviewers’ suggestions (see below). In the new analysis we don’t see that transgenic expression of GCaMP is substantially worse than viral expression. Nonetheless, we have addressed the potential difference where we list and discuss possible causes. Neuropil contamination is one possible explanation and strength of expression may be another.

3) Of course, it is expected that SNR will decrease with zoom at constant power because of shot noise. But it is highly unlikely that investigators would use the same laser power for imaging over changes in zoom by a factor of 20. Please comment.

We used the same laser power when switching between high and low zoom, to enable the most direct comparison possible between these conditions. However, we agree that it’s unlikely investigators would use the same laser power for imaging under such different conditions. In light of this fact, and as requested by reviewer 2 in the initial reviews, we have now replaced the low-zoom results with downsampled data. The baseline noise of the downsampled videos was similar to that in the Allen Brain Observatory (Figure 4), consistent with the downsampled results being representative of real world imaging conditions at low zoom. In the revised manuscript, we compare spike detection for each neuron under high zoom and after downsampling and we place high zoom and downsampled results side-by-side to facilitate comparison (Figures 5-7).

4) Critical references are missing. Dana et al., 2014, and Wei et al., 2019, have shown that `GCaMP6s cells have spike-triggered fluorescence responses of larger amplitude, lower variability and greater single-spike detectability than GCaMP6f` in transgenic mice.

We have cited additional references, including Dana et al., 2014 and Wei et al., 2019.

5) Important spike-to-calcium parameters (e.g. rise- and decay- time constants) are missing. These parameters should be reported as part of the basic analyses; please incorporate the analyses e.g. Figures 2D,E,F and 3F in Chen et al., 2013. One can also use existing spike-to-fluorescence models to estimate the parameters (Wei et al., 2019).

In Figure 3, we now provide examples of fluorescence transients, mean transients, and plots of peak DF/F, rise time constant and decay time constant for 1-5 APs.

6) Spike-event snippet creation. First, the authors should demonstrate how they chose the parameters for the snippets in each imaging condition. It is not clear that the choice of snippet parameters were optimal; for example, Emx1-s and tetO-s ΔF/F at 4APs in Figure 2 seem not to reach the maximum within 200 ms window. Ideally, these parameters could be determined by the estimates of rise- and decay- time constants. For example, if the half-rise time is 100 ms and the spike-event snippet is 200 ms, one should use a time window at least 400 ms to capture the peak dF/F.

We have lengthened the time windows for snippet creation and over which peak ΔF/F was found. The rise and decay time constants for GCaMP6f are 50-100 ms and 200-300 ms; and for GCaMP6s 150-200 ms and ~750 ms (4-5 APs, Figure 3C). Our updated snippets extend 300 ms and 500 ms after the spike, for GCaMP6f and GCaMP6s and are now adequate to robustly capture the peak ΔF/F (Figures 1 and 3).

7) Neuropil removal. This is a confusing part throughout the manuscript. At the beginning, the authors preferred not performing neuropil correction because it might increase peak ΔF/F variability (Figure 2—figure supplement 1D), then in the later part, the authors claimed that the importance of the neuropil correction to the spike detection. The authors could offer a systematic study to address if neuropil should be removed and how. It seems like the optimal r for each cell should be used throughout the manuscript. One study from Kerlin et al., 2010, (this paper should be cited) provided some answers to this.

We have streamlined neuropil subtraction in the revised manuscript. We determined the r-value that optimizes spike detection for GCaMP6f lines (spike detection changes little with r-value in GCaMP6s lines) and found that the mean optimal r-value is 0.82 (Figure 2). For high zoom images, we perform neuropil subtraction for all neurons, throughout the manuscript, with r = 0.8 except where r = 0.8 resulted in negative pre-spike fluorescence. For downsampled images, we used the neuropil subtraction algorithm employed by the Allen Brain Observatory, to mimic population imaging conditions as closely as possible. New methods sections explain how we performed neuropil subtraction.

8) ΔF/F computation. In general one would take a long time-window to compute F0 with the background subtraction in the denominator, where F0 aims to reflect 0AP fluorescence. Computing locally as the mean within 50 or 20 ms before the first spike event is risky, because F0 in a short window can be contaminated by the previous calcium decay after a many-AP event e.g. F0 around 5-AP in Figure 1B.

We’ve revised our ΔF/F calculation to reduce its sensitivity to previous spikes. For each snippet we calculated ΔF by subtracting the pre-spike fluorescence, defined as the mean fluorescence 0-100 ms before the spike. As the denominator in the ΔF/F calculation, we used the minimum pre-spike fluorescence across a group of snippets, with snippets grouped by cell and by number of APs. This approach ensures that the same F0 value is used for all snippets in a group, even where the transients in some snippets ride on the tail of the fluorescence transient from a previous AP. Manual inspection revealed that F0 returned to a stable baseline (presumably 0 AP fluorescence) in many trials and that our updated procedure uses 0 AP fluorescence as the denominator of the ΔF/F calculation for all trials.

9) Peak ΔF/F variability. First, it is not clear how the mean coefficient of variation of ΔF/F peak (a main measure of peak ΔF/F variability throughout the manuscript) was defined and computed. Is the coefficient in the term of the number of spikes? How was it computed in a given nAP case, e.g. Figure 4A right or Figure 4B? Second, although shot noise is dominant on single pixels, or at high sample rates or in low brightness conditions, variability would be reduced when computing F by averaging over N pixels. The analysis on single pixels is somewhat misleading. Third, trial-by-trial variability is important but the measurement is not clear. In general, the peak ΔF/F should depend on the time series of the spikes. It roughly depends on the number of the spikes in a brief time window. One would thus expect two sources of trial-by-trial variance, one depends on the number of spikes; the other depends on the spike pattern as the number of spikes is fixed. Authors should be able to decompose the variability into these terms.10) The finding that variance in response amplitude is no larger than expected from shot-noise is surprising and unlikely to be true. There are many reasons why the coupling between AP and Ca might be variable (modulation, baseline potential, state of channels, channel fluctuations). Make sure this analysis is correct.

We have removed the coefficient of variation measurements form the paper. In light of the reviewers’ concerns about single pixel measurements, we have revised our analysis of noise and trial-to-trial variability, summing pixels across each soma. Our revised analysis indicates that variability was often greater than that expected from shot noise (Figure 3—figure supplement 1). Our results document trial-to-trial variability but provide little further information on sources of variability. Likely motion is not a significant contributor since there’s very little motion in the images. We speculate that biology is the major source of trial-to-trial variability.

11) Lastly, as demonstrated above, the peak ΔF/F does not directly depend on behavioral condition when the spike pattern is given. Yet the difference in spon vs visstim in Cux2-f mice is striking. Can this difference be explained by the spike pattern difference in spon vs visstim conditions? If not, what causes this difference?

The comparison of spon and visstim conditions was based on a small number of Cux2-f neurons. The revision of selection procedures when updating the manuscript has reduced the number of Cux2-f neurons without visual stimuli to zero (Figure 1—figure supplement 1). Whether there’s a difference in peak ΔF/F between spon and visstim in Cux2-f is not a question we can answer with confidence with our dataset.

[Editors' note: further revisions were suggested prior to acceptance, as described below.]We are satisfied with the data in the revised manuscript, but ask that you revise the text in accord with the comments below as we believe that these changes with substantially improve the impact of your important work.Reviewer #2:Far more emphasis needs to be placed on the "low zoom" data. The difference between the mouse lines is modest at best. However, the demonstration that spike detection is ~15% under normal imaging conditions is a stunning observation that should be a wake-up call to large parts of the community.

We have made several changes to emphasize the low detection rate under normal imaging conditions. Main changes include adding AP detection rate numbers to the Abstract. We have also rewritten the Introduction to include a new second paragraph addressing the paradox that APs can be detected efficiently with GCaMP indicators, but often aren’t in practice due to microscope magnification and other constraints such as those on model refinement.

Reviewer #3:The authors have done a great deal to improve the study. There are some remaining points can be easily addressed to improve the study even further.1) There is no description of what cells the Cux2 and Emx1 cre lines target. It would help make the study more approachable and useful to include this information here.

In the Results we now note that expression is in excitatory neurons:

“The dataset for further analysis was from 48 neurons from mice of 4 transgenic lines, two expressing GCaMP6s and two GCaMP6f in excitatory neurons in layer 2/3 and deeper layers of cortex (Table 1).”

We provide additional information on expression in the Materials and methods, where we now state:

“All four lines drive GCaMP expression primarily in excitatory neurons. In Cux2-CreERT2 mice, Cre and GCaMP expression are enriched in layer 2/3 (Franco et al., 2012; Harris et al., 2014). In Emx1-IRES-Cre and Camk2a-tTA mice, GCaMP is expressed throughout cortical layers (Gorski et al., 2002; Wekselblatt et al., 2016). Images showing the pattern of Cre and GCaMP expression in these mouse lines are available via the Transgenic Characterization pages of the Allen Mouse Brain Connectivity Atlas and Allen Brain Observatory: https://connectivity.brain-map.org/transgenic, http://observatory.brain-map.org/visualcoding/transgenic”

2) The authors report that there is no epileptiform activity in the mice they use. Do they not see such activity in these mice in their hands or do they select individual mice without seizures? How to they judge the presence of epileptiform activity?

The Materials and methods previously stated, “Ai93 and Ai94 containing mice included in this dataset did not show behavioral signs for epileptic brain activity.” We have replaced this sentence with a more specific statement:

“Mice of some of the genotypes used here, most notably Emx1-f, can exhibit epileptiform activity (Steinmetz et al., 2017), including overt seizures. Mice with seizures were excluded from the study.”

3) In light of point 2, the fraction of spikes that are in groups of 1-5 spike bursts is very different for GCAMP6s and GCAMP6f mice. This is a metric measured by the cell-attached recording. Doesn't this indicate that the transgene has a large effect on cellular or circuit activity patterns?

We agree with the reviewer’s interpretation and now made this point explicitly in the revised manuscript:

“However, the spiking patterns of neurons from GCaMP6s and -f lines commonly differed, suggesting that one or more transgenes affected cell or circuit activity (Figure 1—figure supplement 1C).”

4) The authors use QC metrics to reduce the number of cells analyzed from >200 to 48. Can they report what factors result in 75% of the cells being rejected? Do they know which rejection factors actually impact the ability to accurately infer spiking from fluorescence? Such insight would be very useful for others who don't have cell-attached recordings but want to be able to understand what cells to include in their final analyses.

Most of the discarded neurons (145 of 213) were removed by the electrophysiology quality control filters. These filters often removed overlapping populations of neurons, making it impossible to assign removal to single factors. The overlapping nature of the filtering process also makes it difficult or perhaps impossible to determine the impact of each of our 35 electrophysiology filters on our ability to infer spiking from fluorescence. A few (10 of 68) neurons were removed during image downsampling, leaving 58 neurons. 10 neurons were removed manually, 7 with red indicator in the soma, 2 because of image processing issues, and 1 because of a truncated electrophysiology recording, leaving 48 neurons in the final data set. To the Materials and methods, we have added information on the number of neurons removed by each step in our quality control process.

5) Lastly, the Discussion leaves something to be desired. It is a synopsis of the conclusions and iterates some rather obvious points but also making some claims without backing. It is not clear that the statements about GFP quantum yield being the limiting factor to further improvement of single-spike detection is correct. The Ca-sensitive fluorophores may not get brighter, but their linearity, DF/F, stability etc may improve. Given the assumption that photon-budget is limiting, the statement that voltage-sensors may provide the solution seems wrong as these will provide many fewer photons per AP. The authors could use the discussion instead of provide some helpful hints to imagers to make the best use of their data based on the analyses presented.

We have rewritten this paragraph of the Discussion, now suggesting best practices in light our results. The revised paragraph reads:

“Our results point to several practices that might be adopted to maximize spike detection. Firstly, minimize the field of view, maximizing photon flux per neuron. Secondly, tune the spike inference model for each neuron independently, where possible. Thirdly, compare the results of several spike inference models. The 3 models employed here produced similar AP detection rates, whether applied to high resolution or to down-sampled images. Similarly, Pachitariu et al., (2018) observed that the L0 constraint failed to improve performance of the NND model. Nonetheless, each model has strengths and weaknesses. For example, a model may detect more APs than another but at the cost of a greater false positive rate. As a result, model performance may diverge for some AP rates and patterns. In the worst case, comparing models provides some protection from errors in implementation. Fourthly, ensure that traces are sampled (or upsampled) at a sufficiently high rate when employing NND and use autocalibration with MLspike; both make a substantial difference to model performance. Finally, exercise caution when interpreting the inferred spike rates. Commonly, many APs are not detected using even the most accurate spike inference models.”